# Adaptive Influence Maximization with Myopic Feedback

**Binghui Peng**\*
Columbia University
bp2601@columbia.edu

**Wei Chen**
Microsoft Research
weic@microsoft.com

## Abstract

We study the *adaptive influence maximization problem with myopic feedback* under the independent cascade model: one sequentially selects $k$ nodes as seeds one by one from a social network, and each selected seed returns the immediate neighbors it activates as the feedback available for later selections, and the goal is to maximize the expected number of total activated nodes, referred as the *influence spread*. We show that the *adaptivity gap*, the ratio between the optimal adaptive influence spread and the optimal non-adaptive influence spread, is at most $4$ and at least $e/(e-1)$, and the approximation ratios with respect to the optimal adaptive influence spread of both the non-adaptive greedy and adaptive greedy algorithms are at least $\frac{1}{4}(1-\frac{1}{e})$ and at most $\frac{e^2+1}{(e+1)^2} < 1 - \frac{1}{e}$. Moreover, the approximation ratio of the non-adaptive greedy algorithm is no worse than that of the adaptive greedy algorithm, when considering all graphs. Our result confirms a long-standing open conjecture of Golovin and Krause (2011) on the constant approximation ratio of adaptive greedy with myopic feedback, and it also suggests that adaptive greedy may not bring much benefit under myopic feedback.

## 1 Introduction

Influence maximization is the task of given a social network and a stochastic diffusion model on the network, finding the $k$ seed nodes with the largest expected influence spread in the model [11]. Influence maximization and its variants have applications in viral marketing, rumor control, etc. and have been extensively studied (cf. [6, 12]).

In this paper, we focus on the *adaptive influence maximization* problem, where seed nodes are sequentially selected one by one, and after each seed selection, partial or full diffusion results from the seed are returned as the feedback, which could be used for subsequent seed selections. Two main types of feedback have been proposed and studied before: (a) *full-adoption feedback*, where the entire diffusion process from the seed selected is returned as the feedback, and (b) *myopic feedback*, where only the immediate neighbors activated by the selected seed are returned as the feedback. Under the common independent cascade (IC) model where every edge in the graph has an independent probability of passing influence, Golovin and Krause [7] show that the full-adoption feedback model satisfies the key *adaptive submodularity* property, which enables a simple adaptive greedy algorithm to achieve a $(1 - 1/e)$ approximation to the adaptive optimal solution. However, the IC model with myopic feedback is not adaptive submodular, and Golovin and Krause [7] only conjecture that in this case the adaptive greedy algorithm still guarantees a constant approximation. To the best of our knowledge, this conjecture is still open before our result in this paper, which confirms that indeed adaptive greedy is a constant approximation of the adaptive optimal solution.

In particular, our paper presents two sets of related results on adaptive influence maximization with myopic feedback under the IC model. We first study the *adaptivity gap* of the problem (Section 3), which is defined as the ratio between the adaptive optimal solution and the non-adaptive optimal solution, and is an indicator on how useful the adaptivity could be to the problem. We show that the adaptivity gap for our problem is at most 4 (Theorem 1) and at least $e/(e-1)$ (Theorem 2). The proof of the upper bound 4 is the most involved, because the problem is not adaptive submodular, and we have to create a hybrid policy that involves three independent runs of the diffusion process in order to connect between an adaptive policy and a non-adaptive policy. Next we study the approximation ratio with respect to the adaptive optimal solution for both non-adaptive greedy and adaptive greedy algorithms (Section 4). We show that the approximation ratios of both algorithms are at least $\frac{1}{4}(1 - \frac{1}{e})$ (Theorem 3), which combines the adaptivity upper bound of 4 with the results that both algorithms achieve $(1 - 1/e)$ approximation of the non-adaptive optimal solution (the $(1 - 1/e)$ approximation ratio for the adaptive greedy algorithm requires a new proof). We further show that the approximation ratios for both algorithms are at most $\frac{e^2+1}{(e+1)^2} \approx 0.606$, which is strictly less than $1 - 1/e \approx 0.632$, and the approximation ratio of non-adaptive greedy is the same as the worst approximation ratio of the adaptive greedy over a family of graphs (Theorem 4).

In summary, our contribution is the systematic study on adaptive influence maximization with myopic feedback under the IC model. We prove both constant upper and lower bounds on the adaptivity gap in this case, and constant upper and lower bounds on the approximation ratios (with respect to the optimal adaptive solution) achieved by non-adaptive greedy and adaptive greedy algorithms. The constant approximation ratio of the adaptive greedy algorithm answers a long-standing open conjecture affirmatively. Our result on the adaptivity gap is the first one on a problem not satisfying adaptive submodularity. Our results also suggest that adaptive greedy may not bring much benefit under the myopic feedback model.

Due to the space constraint, full proof details are included in the supplementary material.

**Related Work.** Influence maximization as a discrete optimization task is first proposed by Kempe et al. [11], who propose the independent cascade, linear threshold and other models, study their submodularity and the greedy approximation algorithm for the influence maximization task. Since then, influence maximization and its variants have been extensively studied. We refer to recent surveys [6, 12] for the general coverage of this area.

Adaptive submodularity is formulated by Golovin and Krause [7] for general stochastic adaptive optimization problems, and they show that the adaptive greedy algorithm achieves $1 - 1/e$ approximation if the problem is adaptive monotone and adaptive submodular. They study the influence maximization problem under the IC model as an application, and prove that the full-adoption feedback under the IC model is adaptive submodular. However, in their arXiv version, they show that the myopic feedback version is not adaptive submodular, and they conjecture that adaptive greedy would still achieve a constant approximation in this case.

Adaptive influence maximization has been studied in [19, 20, 16, 13, 18, 10, 17, 5]. Tong et al. [19] provide both adaptive greedy and efficient heuristic algorithms for adaptive influence maximization. Their theoretical analysis works for the full-adoption feedback model but has a gap when applied to myopic feedback, which is confirmed by the authors. Yuan and Tang [20] introduce the partial feedback model and develop algorithms that balance the tradeoff between delay and performance, and their partial feedback model does not coincide with the myopic feedback model. Salha et al. [13] consider a different diffusion model where edges can be reactivated at each time step, and they show that myopic feedback under this model is adaptive submodular. Sun et al. [16] study the multi-round adaptive influence maximization problem, where $k$ seeds are selected in each round and at the end of the round the full-adoption feedback is returned. Tong [18] introduces a general feedback model and develops some heuristic algorithms for this model. Han et al. [10] and Tang et al. [17] propose efficient adaptive algorithms for influence maximization and seed set minimization respectively based on the reverse influence sampling approach, both for IC models with full-adoption feedback. In a separate paper [5], we study the adaptivity gap in the IC model with full-adoption feedback for several classes of graphs such as trees and bipartite graphs. A different two stage seeding process has also been studied [14, 3, 15], but the model is quite different, since their first stage of selecting a node set $X$ is only to introduce the neighbors of $X$ as seeding candidates for the second stage.

Adaptivity gap has been studied by two lines of research. The first line of work utilizes multilinear extension and adaptive submodularity to study adaptivity gaps for the class of stochastic submodular maximization problems and give an $e/(e-1)$ upper bound for matroid constraints [2, 1]. The second line of work [8, 9, 4] studies the stochastic probing problem and proposes the idea of random-walk non-adaptive policy on the decision tree, which partially inspires our analysis. However, their analysis also implicitly depends on adaptive submodularity. In contrast, our result on the adaptivity gap is the first on a problem that does not satisfy adaptive submodularity (see Section 3.1 for more discussions).

## 2 Model and Problem Definition

**Diffusion Model.** In this paper, we focus on the well known Independent Cascade (IC) model as the diffusion model. In the IC model, the social network is described by a directed influence graph $G = (V, E, p)$, where $V$ is the set of nodes ($|V| = n$), $E \subseteq V \times V$ is the set of directed edges, and each directed edge $(u, v) \in E$ is associated with a probability $p_{uv} \in [0, 1]$. The *live edge* graph $L = (V, L(E))$ is a random subgraph of $G$, for any edge $(u, v) \in E$, $(u, v) \in L(E)$ with independent probability $p_{uv}$. If $(u, v) \in L(E)$, we say edge $(u, v)$ is *live*, otherwise we say it is *blocked*. The dynamic diffusion in the IC model is as follows: at time $t = 0$ a live-edge graph $L$ is sampled and nodes in a seed set $S \subseteq V$ are activated. At every discrete time $t = 1, 2, \ldots$, if a node $u$ was activated at time $t - 1$, then all of $u$'s out-going neighbors in $L$ are activated at time $t$. The propagation continues until there are no more activated nodes at a time step. The dynamic model can be viewed equivalently as every activated node $u$ has one chance to activate each of its out-going neighbor $v$ with independent success probability $p_{uv}$. Given a seed set $S$, the *influence spread* of $S$, denoted $\sigma(S)$, is the expected number of nodes activated in the diffusion process from $S$, i.e. $\sigma(S) = \mathbb{E}_L[|\Gamma(S, L)|]$, where $\Gamma(S, L)$ is the set of nodes reachable from $S$ in graph $L$.

**Influence Maximization Problem.** Under the IC model, we formalize the influence maximization (IM) problem in both non-adaptive and adaptive settings. Influence maximization in the non-adaptive setting follows the classical work of [11], and is defined below.

**Definition 1** (Non-adaptive Influence Maximization). Non-adaptive influence maximization *is the problem of given a directed influence graph $G = (V, E, p)$ with IC model parameters $\{p_{uv}\}_{(u,v) \in E}$ and a budget $k$, finding a seed set $S^*$ of at most $k$ nodes such that the influence spread of $S^*$, $\sigma(S^*)$, is maximized, i.e. finding $S^* \in \text{argmax}_{S \subseteq V, |S| \leq k} \sigma(S)$.*

We formulate influence maximization in the adaptive setting following the framework of [7]. Let $O$ denote the set of states, which informally correspond to the feedback information in the adaptive setting. A *realization* $\phi$ is a function $\phi : V \to O$, such that for $u \in V$, $\phi(u)$ represents the feedback obtained when selecting $u$ as a seed node. In this paper, we focus on the *myopic feedback* model [7], which means the feedback of a node $u$ only contains the status of the out-going edges of $u$ being live or blocked. Informally it means that after selecting a seed we can only see its one step propagation effect as the feedback. The realization $\phi$ then determines the status of every edge in $G$, and thus corresponds to a live-edge graph. As a comparison, the *full-adoption feedback* model [7] is such that for each seed node $u$, the feedback contains the status of every out-going edge of every node $v$ that is reachable from $u$ in a live-edge graph $L$. This means that after selecting a seed $u$, we can see the full cascade from $u$ as the feedback. In the full-adoption feedback case, each realization $\phi$ also corresponds to a unique live-edge graph. Henceforth, we refer to $\phi$ as both a realization and a live-edge graph interchangeably. In the remainder of this section, the terminologies we introduce apply to both feedback models, unless we explicitly point out which feedback model we are discussing.

Let $\mathcal{R}$ denote the set of all realizations. We use $\Phi$ to denote a random realization, following the distribution $\mathcal{P}$ over random live-edge graphs (i.e. each edge $(u, v) \in E$ has an independent probability of $p_{uv}$ to be live in $\Phi$). Given a subset $S$ and a realization $\phi$, we define *influence utility function* $f : 2^V \times \mathcal{R} \to \mathbb{R}^+$ as $f(S, \phi) = |\Gamma(S, \phi)|$, where $\mathbb{R}^+$ is the set of non-negative real numbers. That is, $f(S, \phi)$ is the number of nodes reachable from $S$ in realization (live-edge graph) $\phi$. Then it is clear that influence spread $\sigma(S) = \mathbb{E}_{\Phi \sim \mathcal{P}}[f(S, \Phi)]$.

In the *adaptive influence maximization* problem, we could sequentially select nodes as seeds, and after selecting one seed node, we could obtain its feedback, and use the feedback to guide further seed selections. A *partial realization* $\psi$ maps a subset of nodes in $V$, denoted $\text{dom}(\psi)$ for domain of $\psi$, to their states. Partial realization $\psi$ represents the feedback we could obtain after nodes in $\text{dom}(\psi)$ are selected as seeds. For convenience, we also represent $\psi$ as a relation, i.e., $\psi = \{(u, o) \in V \times O :$

$u \in \text{dom}(\psi), o = \psi(u)\}$. We say that a full realization $\phi$ is *consistent* with a partial realization $\psi$, denoted as $\phi \sim \psi$, if $\phi(u) = \psi(u)$ for every $u \in \text{dom}(\psi)$.

An adaptive policy $\pi$ is a mapping from partial realizations to nodes. Given a partial realization $\psi$, $\pi(\psi)$ represents the next seed node that policy $\pi$ would select when it sees the feedback represented by $\psi$. Under a full realization $\phi$ consistent with $\psi$, after selecting $\pi(\psi)$, the policy would obtain feedback $\phi(\pi(\psi))$, and the partial realization would grow to $\psi' = \psi \cup \{(\pi(\psi), \phi(\pi(\psi)))\}$, and policy $\pi$ could pick the next seed node $\pi(\psi')$ based on partial realization $\psi'$. For convenience, we only consider deterministic policies in this paper, and the results we derive can be easily extend to randomized policies. Let $V(\pi, \phi)$ denote the set of nodes selected by policy $\pi$ under realization $\phi$. For the adaptive influence maximization problem, we consider the simple cardinality constraint such that $|V(\pi, \phi)| \leq k$, i.e. the policy only selects at most $k$ nodes. Let $\Pi(k)$ denote the set of such policies.

The objective of an adaptive policy $\pi$ is its *adaptive influence spread*, which is the expected number of nodes that are activated under policy $\pi$. Formally, we define the adaptive influence spread of $\pi$ as $\sigma(\pi) = \mathbb{E}_{\Phi \sim \mathcal{P}}[f(V(\pi, \Phi), \Phi)]$. The adaptive influence maximization problem is defined as follows.

**Definition 2** (Adaptive Influence Maximization). Adaptive influence maximization *is the problem of given a directed influence graph $G = (V, E, p)$ with IC model parameters $\{p_{uv}\}_{(u,v) \in E}$ and a budget $k$, finding an adaptive policy $\pi^*$ that selects at most $k$ seed nodes such that the adaptive influence spread of $\pi^*$, $\sigma(\pi^*)$, is maximized, i.e. finding $\pi^* \in \text{argmax}_{\pi \in \Pi(k)} \sigma(\pi)$.*

Note that for any fixed seed set $S$, we can create a policy $\pi_S$ that always selects set $S$ regardless of the feedback, which means any non-adaptive solution is a feasible solution for adaptive influence maximization. Therefore, the optimal adaptive influence spread should be at least as good as the optimal non-adaptive influence spread, under the same budget constraint.

**Adaptivity Gap.** Since the adaptive policy is usually hard to design and analyze and the adaptive interaction process may also be slow in practice, a fundamental question for adaptive stochastic optimization problems is whether adaptive algorithms are really superior to non-adaptive algorithms. The *adaptivity gap* measures the gap between the optimal adaptive solution and the optimal non-adaptive solution. More concretely, if we use $\text{OPT}_N(G, k)$ (resp. $\text{OPT}_A(G, k)$) to denote the influence spread of the optimal non-adaptive (resp. adaptive) solution for the IM problem in an influence graph $G$ under the IC model with seed budget $k$, then we have the following definition.

**Definition 3** (Adaptivity Gap for IM). *The adaptivity gap in the IC model is defined as the supremum of the ratios of the influence spread between the optimal adaptive policy and the optimal non-adaptive policy, over all possible influence graphs $G$ and seed budget $k$, i.e.,*

$$\sup_{G,k} \frac{\text{OPT}_A(G, k)}{\text{OPT}_N(G, k)}. \tag{1}$$

**Submodularity and Adaptive Submodularity.** Non-adaptive influence maximization is often solved via submodular function maximization technique. A set function $f : 2^V \to \mathbb{R}$ is *submodular* if for all $S \subseteq T \subseteq V$ and all $u \in V \setminus T$, $f(S \cup \{u\}) - f(S) \geq f(T \cup \{u\}) - f(T)$. Set function $f$ is monotone if for all $S \subseteq T \subseteq V$, $f(S) \leq f(T)$. Kempe et al. [11] show that the influence spread function $\sigma(S)$ under the IC model is monotone and submodular, and thus a simple non-adaptive greedy algorithm achieves a $(1 - \frac{1}{e})$ approximation of the optimal non-adaptive solution, assuming function evaluation $\sigma(S)$ is given by an oracle.

Golovin and Krause [7] define *adaptive submodularity* for the adaptive stochastic optimization framework. In the context of adaptive influence maximization, adaptive submodularity can be defined as follows. Given a utility function $f$, for any partial realization $\psi$ and a node $u \notin \text{dom}(\psi)$, we define the marginal gain of $u$ given $\psi$ as $\Delta_f(u \mid \psi) = \mathbb{E}_{\Phi \sim \mathcal{P}}[f(\text{dom}(\psi) \cup \{u\}, \Phi) - f(\text{dom}(\psi), \Phi) \mid \Phi \sim \psi]$, i.e. the expected marginal gain on influence spread when adding $u$ to the partial realization $\psi$. A partial realization $\psi$ is a *sub-realization* of another partial realization $\psi'$ if $\psi \subseteq \psi'$ when treating both as relations. We say that the utility function $f$ is *adaptive submodular* with respect to $\mathcal{P}$ if for any two fixed partial realizations $\psi$ and $\psi'$ such that $\psi \subseteq \psi'$, for any $u \notin \text{dom}(\psi')$, we have $\Delta_f(u \mid \psi) \geq \Delta_f(u \mid \psi')$, that is, the marginal influence spread of a node given more feedback is at most its marginal influence spread given less feedback. We say that $f$ is *adaptive monotone* with respect to $\mathcal{P}$ if for any partial realization $\psi$ with $\text{Pr}_{\Phi \sim \mathcal{P}}(\Phi \sim \psi) > 0$, $\Delta_f(u \mid \psi) \geq 0$.

Golovin and Krause [7] show that the influence utility function under the IC model with full-adoption feedback is adaptive monotone and adaptive submodular, and thus the adaptive greedy algorithm achieves $(1 - \frac{1}{e})$ approximation of the adaptive optimal solution. However, they show that the influence utility function under the IC model with myopic feedback is not adaptive submodular. They conjecture that the adaptive greedy policy still provides a constant approximation. In this paper, we show that the adaptive greedy policy provides a $\frac{1}{4}(1 - \frac{1}{e})$ approximation, and thus finally address this conjecture affirmatively.

# 3 Adaptivity Gap in Myopic Feedback Model

In this section, we analyze the adaptivity gap for influence maximization problems under the myopic feedback model and derive both upper and lower bounds.

## 3.1 Upper Bound on the Adaptivity Gap

Our main result is an upper bound on the adaptivity gap for the myopic feedback model, which is formally stated below.

**Theorem 1.** *Under the IC model with myopic feedback, the adaptivity gap for the influence maximization problem is at most 4.*

**Proof outline.** We now outline the main ideas and the structure of the proof of Theorem 1. The main idea is to show that for each adaptive policy $\pi$, we could construct a non-adaptive randomized policy $\mathcal{W}(\pi)$, such that the adaptive influence spread $\sigma(\pi)$ is at most four times the non-adaptive influence spread of $\mathcal{W}(\pi)$, denoted $\sigma(\mathcal{W}(\pi))$. This would immediately imply Theorem 1. The non-adaptive policy $\mathcal{W}(\pi)$ is constructed by viewing adaptive policy $\pi$ as a decision tree with leaves representing the final seed set selected (Definition 4), and $\mathcal{W}(\pi)$ simply samples such a seed set based on the distribution of the leaves (Definition 5). The key to connect $\sigma(\pi)$ with $\sigma(\mathcal{W}(\pi))$ is by introducing a fictitious hybrid policy $\bar{\pi}$, such that $\sigma(\pi) \leq \bar{\sigma}(\bar{\pi}) \leq 4\sigma(\mathcal{W}(\pi))$, where $\bar{\sigma}(\bar{\pi})$ is the *aggregate adaptive influence spread* (defined in Eqs. (2) and (3)). Intuitively, $\bar{\pi}$ works on three independent realizations $\Phi^1, \Phi^2, \Phi^3$ and it adaptively selects seeds as $\pi$ working on $\Phi^1$. The difference is that each selected seed has three independent chances to activate its out-neighbors according to the union of $\Phi^1, \Phi^2, \Phi^3$. The inequality $\sigma(\pi) \leq \bar{\sigma}(\bar{\pi})$ is immediate and the main effort is on proving $\bar{\sigma}(\bar{\pi}) \leq 4\sigma(\mathcal{W}(\pi))$.

To do so, we first introduce general notations $\sigma^t(S)$ and $\sigma^t(\pi)$ with $t = 1, 2, 3$, where $\sigma^t(S)$ is the *t-th aggregate influence spread* for a seed set $S$ and $\sigma^t(\pi)$ is the *t-th aggregate adaptive influence spread* for an adaptive policy $\pi$, and they mean that all seed nodes have $t$ independent chances to activate their out-neighbors. Obviously, $\bar{\sigma}(\bar{\pi}) = \sigma^3(\pi)$ and $\sigma(\mathcal{W}(\pi)) = \sigma^1(\mathcal{W}(\pi))$. We then represent $\sigma^t(\mathcal{W}(\pi))$ and $\sigma^t(\pi)$ as a summation of $k$ non-adaptive marginal gains $\Delta_{f^t}(u \mid \text{dom}(\psi_s))$'s and adaptive marginal gains $\Delta_{f^t}(u \mid \psi_s)$'s, respectively (Definition 6 and Lemma 1), with respect to node $s$ in different levels of the decision tree. Next, we establish the key connection between the adaptive marginal gain and the nonadaptive marginal gain (Lemma 3): $\Delta_{f^3}(u \mid \psi^1) \leq 2\Delta_{f^2}(u \mid \text{dom}(\psi^1))$. This immediately implies that $\sigma^3(\pi) \leq 2\sigma^2(\mathcal{W}(\pi))$. Finally, we prove that the $t$-th aggregate non-adaptive influence spread $\sigma^t(S)$ is bounded by $t \cdot \sigma(S)$, which implies that $\sigma^2(\mathcal{W}(\pi)) \leq 2\sigma(\mathcal{W}(\pi))$. This concludes the proof.

We remark that our introduction of the hybrid policy $\bar{\pi}$ is inspired by the analysis in [4], which shows that the adaptivity gap for the *stochastic multi-value probing (SMP)* problem is at most 2. However, our analysis is more complicated than theirs and thus is novel in several aspects. First, the SMP problem is simpler than our problem, with the key difference being that SMP is adaptive submodular but our problem is not. Therefore, we cannot apply their way of inductive reasoning that implicitly relies on adaptive submodularity. Instead, we have to use our marginal gain representation and redo the bounding analysis carefully based on the (non-adaptive) submodularity of the influence utility function on live-edge graphs. Moreover, our influence utility function is also sophisticated and we have to use three independent realizations in order to apply the submodularity on live-edge graphs, which results in an adaptivity bound of 4, while their analysis only needs two independent realizations to achieve a bound of 2. We now provide the technical proof of Theorem 1. We first formally define the decision tree representation.

**Definition 4** (Decision tree representation for adaptive policy). *An adaptive policy $\pi$ can be seen as a decision tree $\mathcal{T}(\pi)$, where each node $s$ of $\mathcal{T}(\pi)$ corresponds to a partial realization $\psi_s$, with the root being the empty partial realization, and node $s'$ is a child of $s$ if $\psi_{s'} = \psi_s \cup \{(\pi(\psi_s), \phi(\pi(\psi_s)))\}$ for some realization $\phi \sim \psi_s$. Each node $s$ is associated with a probability $p_s$, which is the probability that the policy $\pi$ generates partial realization $\psi_s$, i.e. the probability that the policy would walk on the tree from the root to node $s$.*

Next we define the non-adaptive randomized policy $\mathcal{W}(\pi)$, which randomly selects a leaf of $\mathcal{T}(\pi)$.

**Definition 5** (Random-walk non-adaptive policy [9]). *For any adaptive policy $\pi$, let $\mathcal{L}(\pi)$ denote the set of leaves of $\mathcal{T}(\pi)$. Then we construct a randomized non-adaptive policy $\mathcal{W}(\pi)$ as follows: for any leaf $\ell \in \mathcal{L}(\pi)$, $\mathcal{W}(\pi)$ picks leaf $\ell$ with probability $p_\ell$ and selects $\mathrm{dom}(\psi_\ell)$ as the seed set.*

Before proceeding further with our analysis, we introduce some notations for the myopic feedback model. In the myopic feedback model, we notice that the state spaces for all nodes are mutually independent and disjoint. Thus we could decompose the realization space $\mathcal{R}$ into independent subspace, $\mathcal{R} = \times_{u \in V} O_u$, where $O_u$ is the set of all possible states for node $u$. For any full realization $\phi$ (resp. partial realization $\psi$), we would use $\phi_S$ (resp. $\psi_S$) to denote the feedback for the node set $S \subseteq V$. Note that $\phi_S$ and $\psi_S$ are partial realizations with domain $S$. Similarly, we would also use $\mathcal{P}_S$ to denote the probability space $\times_{u \in S} \mathcal{P}_u$, where $\mathcal{P}_u$ is the probability distribution over $O_u$ (i.e. each out-going edge $(u, v)$ of $u$ is live with independent probability $p_{uv}$). With a slight abuse of notation, we further use $\phi_S$ (resp. $\psi_S$) to denote the set of live edges leaving from $S$ under $\phi$ (resp. $\psi$). Then we could use notation $\phi_S^1 \cup \phi_S^2$ to represent the union of live-edges from $\phi^1$ and $\phi^2$ leaving from $S$, and similarly $\psi \cup \phi_S^2$ with $\mathrm{dom}(\psi) = S$.

**Construction of the hybrid policy $\bar{\pi}$.** For any adaptive policy $\pi$, we define a fictitious hybrid policy $\bar{\pi}$ that works on three independent random realizations $\Phi^1$, $\Phi^2$ and $\Phi^3$ simultaneously, thinking about them as from three copies of the graphs $G_1$, $G_2$ and $G_3$. Note that $\bar{\pi}$ is not a real adaptive policy — it is only used for our analytical purpose to build connections between the adaptive policy $\pi$ and the non-adaptive policy $\mathcal{W}(\pi)$. In terms of adaptive seed selection, $\bar{\pi}$ acts exactly the same as $\pi$ on $G_1$, responding to partial realizations $\psi^1$ obtained so far from the full realization $\Phi^1$ of $G_1$, and disregarding the realizations $\Phi^2$ and $\Phi^3$. However, the difference is when we define adaptive influence spread for $\bar{\pi}$, we aggregate the three partial realizations on the seed set together. More precisely, for any $t = 1, 2, 3$, we define the $t$-th aggregate influence utility function as $f^t : 2^V \times \mathcal{R}^t \to \mathbb{R}^+$

$$f^t\left(S, \phi^1, \cdots, \phi^t\right) := f\left(S, (\cup_{i \in [t]} \phi_S^i, \phi_{V \setminus S}^1)\right), \tag{2}$$

where $(\cup_{i \in [t]} \phi_S^i, \phi_{V \setminus S}^1)$ means a new realization $\phi'$ where on set $S$ its set of out-going live-edges is the same as the union of $\phi^1, \cdots \phi^t$, and on set $V \setminus S$ its set of out-going live-edges is the same as $\phi^1$, and $f$ is the original influence utility function defined in Section 2. The objective of the hybrid policy $\bar{\pi}$ is then defined as the adaptive influence spread under policy $\bar{\pi}$, i.e.,

$$\bar{\sigma}(\bar{\pi}) := \mathop{\mathbb{E}}_{\Phi^1, \Phi^2, \Phi^3 \sim \mathcal{P}}\left[f^3(V(\pi, \Phi^1), \Phi^1, \Phi^2, \Phi^3)\right]$$

$$= \mathop{\mathbb{E}}_{\Phi^1, \Phi^2, \Phi^3 \sim \mathcal{P}}\left[f\left(V(\pi, \Phi^1), (\Phi_{V(\pi, \Phi^1)}^1 \cup \Phi_{V(\pi, \Phi^1)}^2 \cup \Phi_{V(\pi, \Phi^1)}^3, \Phi_{V \setminus V(\pi, \Phi^1)}^1)\right)\right]. \tag{3}$$

In other words, the adaptive influence spread of the hybrid policy $\bar{\pi}$ is the influence spread of seed nodes $V(\pi, \Phi^1)$ selected in graph $G_1$ by policy $\pi$, where the live-edge graph on the seed set part $V(\pi, \Phi^1)$ is the union of live-edge graphs of $G_1$, $G_2$ and $G_3$, and the live-edge graph on the non-seed set part is only that of $G_1$. It can also be viewed as each seed node has three independent chances to activate its out-neighbors. Since the hybrid policy $\bar{\pi}$ acts the same as policy $\pi$ on influence graph $G_1$, we can easily conclude:

**Claim 1.** $\bar{\sigma}(\bar{\pi}) \geq \sigma(\pi)$.

We also define $t$-th aggregate influence spread for a seed set $S$, $\sigma^t(S)$, as $\sigma^t(S) = \mathbb{E}_{\Phi^1, \cdots, \Phi^t \sim \mathcal{P}}\left[f^t(S, \Phi^1, \cdots, \Phi^t)\right]$. Then, for the random-walk non-adaptive policy $\mathcal{W}(\pi)$, we define $\sigma^t(\mathcal{W}(\pi)) = \sum_{\ell \in \mathcal{L}(\pi)} p_\ell \cdot \sigma^t(\mathrm{dom}(\psi_\ell))$, that is, the $t$-th aggregate influence spread of $\mathcal{W}(\pi)$ is the average $t$-th aggregate influence spread of seed nodes selected by $\mathcal{W}(\pi)$ according to distribution of the leaves in the decision tree $\mathcal{T}(\pi)$. Similarly, we define the $t$-th aggregate adaptive influence

spread for an adaptive policy $\pi$ as $\sigma^t(\pi) = \mathbb{E}_{\Phi^1,\cdots,\Phi^t \sim \mathcal{P}} \left[ f^t(V(\pi, \Phi^1), \Phi^1, \cdots, \Phi^t) \right]$. Note that $\bar{\sigma}(\bar{\pi}) = \sigma^3(\pi)$.

Now, we could give a definition for the conditional expected marginal gain for the aggregate influence utility function $f^t$ over live-edge graph distributions.

**Definition 6.** *The expected non-adaptive marginal gain of $u$ given set $S$ under $f^t$ is defined as:*

$$\Delta_{f^t}(u \mid S) = \mathop{\mathbb{E}}_{\Phi^1,\cdots,\Phi^t \sim \mathcal{P}} \left[ f^t \left( S \cup \{u\}, \Phi^1, \cdots, \Phi^t \right) - f^t \left( S, \Phi^1, \cdots, \Phi^t \right) \right]. \tag{4}$$

*The expected adaptive marginal gain of $u$ given partial realization $\psi^1$ under $f^t$ is defined as:*

$$\Delta_{f^t}(u \mid \psi^1) = \mathop{\mathbb{E}}_{\Phi^1,\cdots,\Phi^t \sim \mathcal{P}} \left[ f^t \left( \mathrm{dom}(\psi^1) \cup \{u\}, \Phi^1, \cdots, \Phi^t \right) - f^t \left( \mathrm{dom}(\psi^1), \Phi^1, \cdots, \Phi^t \right) \mid \Phi^1 \sim \psi^1 \right]. \tag{5}$$

The following lemma connects $\sigma^t(\pi)$ (and thus $\bar{\sigma}(\bar{\pi})$) with adaptive marginal gain $\Delta_{f^t}(u \mid \psi)$, and connects $\sigma^t(\mathcal{W}(\pi))$ with non-adaptive marginal gain $\Delta_{f^t}(u \mid S)$. Let $\mathcal{P}_i^\pi$ denote the probability distribution over nodes at depth $i$ of the decision $\mathcal{T}(\pi)$. The proof is by applying telescoping series to convert influence spread into the sum of marginal gains.

**Lemma 1.** *For any adaptive policy $\pi$, and $t \geq 1$, we have*

$$\sigma^t(\pi) = \sum_{i=0}^{k-1} \mathop{\mathbb{E}}_{s \sim \mathcal{P}_i^\pi} \left[ \Delta_{f^t} \left( \pi(\psi_s) \mid \psi_s \right) \right], \text{ and } \sigma^t(\mathcal{W}(\pi)) = \sum_{i=0}^{k-1} \mathop{\mathbb{E}}_{s \sim \mathcal{P}_i^\pi} \left[ \Delta_{f^t} \left( \pi(\psi_s) \mid \mathrm{dom}(\psi_s) \right) \right].$$

The next lemma bounds two intermediate adaptive marginal gains to be used for Lemma 3. The proof crucially depend on (a) the independence of realizations $\Phi^1, \Phi^2, \Phi^3$, (b) the independence of feedback of different selected seed nodes, and (c) the submodularity of the influence utility function on live-edge graphs.

**Lemma 2.** *Let $S = \mathrm{dom}(\psi^1)$ and $S^+ = S \cup \{u\}$ for any partial realization $\psi^1$ and any $u \notin \mathrm{dom}(\psi^1)$. Then we have*

$$\mathop{\mathbb{E}}_{\Phi^1,\Phi^2,\Phi^3 \sim \mathcal{P}} \left[ f \left( S^+, (\Phi_S^1 \cup \Phi_S^2 \cup \Phi_S^3, \Phi_u^1 \cup \Phi_u^2, \Phi_{V \setminus S^+}^1) \right) \right.$$
$$\left. - f \left( S, (\Phi_S^1 \cup \Phi_S^2 \cup \Phi_S^3, \Phi_{V \setminus S}^1) \right) \mid \Phi^1 \sim \psi^1 \right] \leq \Delta_{f^2}(u \mid S). \tag{6}$$

$$\mathop{\mathbb{E}}_{\Phi^1,\Phi^2,\Phi^3 \sim \mathcal{P}} \left[ f \left( S^+, (\Phi_S^1 \cup \Phi_S^2 \cup \Phi_S^3, \Phi_u^1 \cup \Phi_u^2 \cup \Phi_u^3, \Phi_{V \setminus S^+}^1) \right) \right.$$
$$\left. - f \left( S^+, (\Phi_S^1 \cup \Phi_S^2 \cup \Phi_S^3, \Phi_u^1 \cup \Phi_u^2, \Phi_{V \setminus S^+}^1) \right) \mid \Phi^1 \sim \psi^1 \right] \leq \Delta_{f^2}(u \mid S). \tag{7}$$

Combining the two inequalities above, we obtain the following key lemma, which bounds the adaptive marginal gain $\Delta_{f^3}(u \mid \psi^1)$ with the non-adaptive marginal gain $\Delta_{f^2}(u \mid \mathrm{dom}(\psi^1))$.

**Lemma 3.** *For any partial realization $\psi^1$ and node $u \notin \mathrm{dom}(\psi^1)$, we have*

$$\Delta_{f^3}(u \mid \psi^1) \leq 2\Delta_{f^2}(u \mid \mathrm{dom}(\psi^1)). \tag{8}$$

The next lemma gives an upper bound on the $t$-th aggregate (non-adaptive) influence spread $\sigma^t(S)$ using the original influence spread $\sigma(S)$. The idea of the proof is that each seed node in $S$ has $t$ independent chances to active its out-neighbors, but afterwards the diffusion is among nodes not in $S$ as in the original diffusion.

**Lemma 4.** *For any $t \geq 1$ and any subset $S \subseteq V$, $\sigma^t(S) \leq t \cdot \sigma(S)$.*

*Proof of Theorem 1.* It is enough to show that for every adaptive policy $\pi$, $\sigma(\pi) \leq 4\sigma(\mathcal{W}(\pi))$. This is done by the following derivation sequence: $\sigma(\pi) \leq \bar{\sigma}(\bar{\pi}) = \sigma^3(\pi) = \sum_{i=0}^{k-1} \mathbb{E}_{s \in \mathcal{P}_i^\pi} \left[ \Delta_{f^3} \left( \pi(\psi_s) \mid \psi_s \right) \right] \leq \sum_{i=0}^{k-1} \mathbb{E}_{s \in \mathcal{P}_i^\pi} \left[ 2\Delta_{f^2} \left( \pi(\psi_s) \mid \mathrm{dom}(\psi_s) \right) \right] = 2\sigma^2(\mathcal{W}(\pi)) \leq 4\sigma(\mathcal{W}(\pi))$, where the first inequality is by Claim 1, the second and the third equalities are by Lemma 1, the second inequality is by Lemma 3 and the last inequality is by Lemma 4. $\square$

## 3.2 Lower bound

Next, we proceed to give a lower bound on the adaptivity gap for the influence maximization problem in the myopic feedback model. Our result is stated as follow:

**Theorem 2.** *Under the IC model with myopic feedback, the adaptivity gap for the influence maximization problem is at least $e/(e-1)$.*

*Proof Sketch.* We construct a bipartite graph $G = (L, R, E, p)$ with $|L| = \binom{m^3}{m^2}$ and $|R| = m^3$. For each subset $X \subset R$ with $|X| = m^2$, there is exactly one node $u \in L$ that connects to all nodes in $X$. We show that for any $\epsilon > 0$, there is a large enough $m$ such that in the above graph with parameter $m$ the adaptivity gap is at least $e/(e-1) - \epsilon$. $\square$

# 4 Adaptive and Non-Adaptive Greedy Algorithms

In this section, we consider two prevalent algorithms — the *greedy* algorithm and the *adaptive greedy* algorithm for the influence maximization problem. To the best of our knowledge, we provide the first approximation ratio of these algorithms with respect to the adaptive optimal solution in the IC model with myopic feedback. We formally describe the algorithms in Figure 1.

| **Greedy Algorithm:** | **Adaptive Greedy Algorithm:** |
|---|---|
| $S = \emptyset$ | $S = \emptyset, \Psi = \emptyset$ |
| **while** $\|S\| < k$ **do** | **while** $\|S\| < k$ **do** |
| $\quad u = \operatorname{argmax}_{u \in V \setminus S} \Delta_f(u\|S)$ | $\quad u = \operatorname{argmax}_{u \in V \setminus S} \Delta_f(u\|\Psi)$ |
| $\quad S = S \cup \{u\}$ | $\quad$ Select $u$ as seed and observe $\Phi(u)$. |
| **end while** | $\quad S = S \cup \{u\}, \Psi = \Psi \cup \{(u, \Phi(u))\}$ |
| **return** $S$ | **end while** |

Figure 1: Description for *greedy* and *adaptive greedy*.

Our main result is summarized below.

**Theorem 3.** *Both greedy and adaptive greedy are $\frac{1}{4}(1 - \frac{1}{e})$ approximate to the optimal adaptive policy under the IC model with myopic feedback.*

*Proof Sketch.* The proof for the non-adaptive greedy algorithm is straightforward since the non-adaptive greedy algorithm provides a $(1 - \frac{1}{e})$ approximation to the non-adaptive optimal solution, which by Theorem 1 is at least $\frac{1}{4}$ of the adaptive optimal solution. For the adaptive greedy algorithm, we need to separately prove that it also provides a $(1 - \frac{1}{e})$ approximation to the non-adaptive optimal solution, and then the result is immediate similar to the non-adaptive greedy algorithm. $\square$

Theorem 3 shows that greedy and adaptive greedy can achieve at least an approximation ratio of $\frac{1}{4}(1 - \frac{1}{e})$ with respect to the adaptive optimal solution. We further show that their approximation ratio is at most $\frac{e^2+1}{(e+1)^2} \approx 0.606$, which is strictly less than $1 - 1/e \approx 0.632$. To do so, we first present an example for non-adaptive greedy with approximation ratio at most $\frac{e^2+1}{(e+1)^2}$. Next, we show that myopic feedback does not help much to adaptive greedy, in that the approximation ratio for the non-adaptive greedy algorithm is no worse than that of adaptive greedy over a family of graphs.

**Theorem 4.** *The approximation ratio for greedy and adaptive greedy is no better than $\frac{e^2+1}{(e+1)^2} \approx 0.606$, which is strictly less than $1 - 1/e \approx 0.632$. Moreover, the approximation ratio of non-adaptive greedy given any influence graph $G$ and budget $k$ is the same as the infimum of the approximation ratios of adaptive greedy on a family of graphs with the same budget $k$.*

# 5  Conclusion and Future Work

In this paper, we systematically study the adaptive influence maximization problem with myopic feedback under the independent cascade model, and provide constant upper and lower bounds on the adaptivity gap and the approximation ratios of the non-adaptive greedy and adaptive greedy algorithms. There are a number of future directions to continue this line of research. First, there is still a gap between the upper and lower bound results in this paper, and thus how to close this gap is the next challenge. Second, our result suggests that adaptive greedy may not bring much benefit under the myopic feedback model, so are there other adaptive algorithms that could do much better? Third, for the IC model with full-adoption feedback, because the feedback on different seed nodes may be correlated, existing adaptivity gap results in [1, 4] cannot be applied even though it is adaptive submodular. For this, our recent study in [5] provides partial answers on several special classes of graphs such as trees and bipartite graphs, but the adaptivity gap on general graphs is still open. One may also explore beyond the IC model, and study adaptive solutions for other models such as the linear threshold model and the general threshold model [11].

## Acknowledgment

Wei Chen is partially supported by the National Natural Science Foundation of China (Grant No. 61433014).

## Footnotes

\*Work is mostly done while Binghui was at Tsinghua University and visiting Microsoft Research Asia as an intern.

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
