[Supplementary Material · myopic_adaptive.pdf]

# Adaptive Influence Maximization with Myopic Feedback

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

 adaptive greedy, when considering over all graphs. Combining with the first observation, we also achieve the result for the adaptive greedy algorithm.

**Theorem 4.** *The approximation ratio for greedy and adaptive greedy is no better than* $\frac{e^2+1}{(e+1)^2} \approx$ *0.606, which is strictly less than* $1 - 1/e \approx 0.632$. *Moreover, the approximation ratio of adaptive greedy is at most that of the non-adaptive greedy, when considering all influence graphs.*

## 5 Conclusion and Future Work

In this paper, we systematically study the adaptive influence maximization problem with myopic feedback under the independent cascade model, and provide constant upper and lower bounds on the adaptivity gap and the approximation ratios of the non-adaptive greedy and adaptive greedy algorithms. There are a number of future directions to continue this line of research. First, there is still a gap between the upper and lower bound results in this paper, and thus how to close this gap is the next challenge. Second, our result suggests that adaptive greedy may not bring much benefit under the myopic feedback model, so are there other adaptive algorithms that could do much better? Third, for the IC model with full-adoption feedback, because the feedback on different seed nodes may be correlated, existing adaptivity gap results in [1, 4] cannot be applied, and thus its adaptivity gap is still open even though it is adaptive submodular. One may also explore beyond the IC model, and study adaptive solutions for other models such as the linear threshold model, general threshold model etc.[9]. Finally, scalable algorithms for adaptive influence maximization is also worth to investigate.

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

# Appendix

We include the missing proofs in this appendix. For convenience, we restate the lemmas and theorems that we prove here.

## A  Missing Proofs of Section 3.1, Adaptivity Upper Bound

**Lemma 1.** *For any adaptive policy $\pi$, and $t \geq 1$, we have*

$$\sigma^t(\pi) = \sum_{i=0}^{k-1} \mathbb{E}_{s \sim \mathcal{P}_i^\pi} \left[ \Delta_{f^t} \left( \pi(\psi_s) \mid \psi_s \right) \right], \text{ and } \sigma^t(\mathcal{W}(\pi)) = \sum_{i=0}^{k-1} \mathbb{E}_{s \sim \mathcal{P}_i^\pi} \left[ \Delta_{f^t} \left( \pi(\psi_s) \mid \mathrm{dom}(\psi_s) \right) \right].$$

*Proof.* We first prove the equality on $\sigma^t(\pi)$. Let $V(\pi, \Phi)_{:i}$ (resp. $V(\pi, \Phi)_i$) denote the first $i$ nodes (resp. the $i^{th}$ node) selected by policy $\pi$ under realization $\Phi$.

Then we have

$$\sum_{i=0}^{k-1} \mathbb{E}_{s \sim \mathcal{P}_i^\pi} \left[ \Delta_{f^t} \left( \pi(\psi_s) \mid \psi_s \right) \right]$$

$$= \sum_{i=0}^{k-1} \mathbb{E}_{s \sim \mathcal{P}_i^\pi} \left[ \mathbb{E}_{\Phi^1, \cdots, \Phi^t \sim \mathcal{P}} \left[ f^t \left( \mathrm{dom}(\psi_s) \cup \pi(\psi_s), \Phi^1, \cdots, \Phi^t \right) - f^t \left( \mathrm{dom}(\psi_s), \Phi^1, \cdots, \Phi^t \right) \mid \Phi^1 \sim \psi_s \right] \right]$$

$$= \sum_{i=0}^{k-1} \mathbb{E}_{\Phi^2, \cdots, \Phi^t \sim \mathcal{P}} \left[ \mathbb{E}_{s \sim \mathcal{P}_i^\pi} \left[ \mathbb{E}_{\Phi^1 \sim \mathcal{P}} \left[ f^t \left( \mathrm{dom}(\psi_s) \cup \pi(\psi_s), \Phi^1, \cdots, \Phi^t \right) - f^t \left( \mathrm{dom}(\psi_s), \Phi^1, \cdots, \Phi^t \right) \mid \Phi^1 \sim \psi_s \right] \right] \right]$$

$$= \sum_{i=0}^{k-1} \mathbb{E}_{\Phi^2, \cdots, \Phi^t \sim \mathcal{P}} \left[ \mathbb{E}_{\Phi^1 \sim \mathcal{P}} \left[ \left( f^t \left( V(\pi, \Phi^1)_{:i} \cup V(\pi, \Phi^1)_{i+1}, \Phi^1, \cdots, \Phi^t \right) - f^t \left( V(\pi, \Phi^1)_{:i}, \Phi^1, \cdots, \Phi^t \right) \right) \right] \right]$$

$$= \mathbb{E}_{\Phi^2, \cdots, \Phi^t \sim \mathcal{P}} \left[ \mathbb{E}_{\Phi^1 \sim \mathcal{P}} \left[ \sum_{i=0}^{k-1} \left( f^t \left( V(\pi, \Phi^1)_{:i} \cup V(\pi, \Phi^1)_{i+1}, \Phi^1, \cdots, \Phi^t \right) - f^t \left( V(\pi, \Phi^1)_{:i}, \Phi^1, \cdots, \Phi^t \right) \right) \right] \right]$$

$$= \mathbb{E}_{\Phi^2, \cdots, \Phi^t \sim \mathcal{P}} \left[ \mathbb{E}_{\Phi^1 \sim \mathcal{P}} \left[ f^t(V(\pi, \Phi^1), \Phi^1, \cdots, \Phi^t) \right] \right]$$

$$= \sigma^t(\pi)$$

The third equality above is by the law of total expectation, and notice that for any tree node $s$ in $\mathcal{T}(\pi)$ and any random realization $\Phi \sim \psi_s$, we have $V(\pi, \Phi)_{:i} = \mathrm{dom}(\psi_s)$ and $V(\pi, \Phi)_{i+1} = \pi(\psi_s)$.

Next, we prove the equality on $\sigma^t(\mathcal{W}(\pi))$.

$$\sum_{i=0}^{k-1} \mathbb{E}_{s \sim \mathcal{P}_i^\pi} \left[ \Delta_{f^t} \left( \pi(\psi_s) \mid \mathrm{dom}(\psi_s) \right) \right]$$

$$= \sum_{i=0}^{k-1} \mathbb{E}_{s \sim \mathcal{P}_i^\pi} \left[ \mathbb{E}_{\Phi^1, \cdots, \Phi^t \sim \mathcal{P}} \left[ f^t \left( \mathrm{dom}(\psi_s) \cup \pi(\psi_s), \Phi^1, \cdots, \Phi^t \right) - f^t \left( \mathrm{dom}(\psi_s), \Phi^1, \cdots, \Phi^t \right) \right] \right]$$

$$= \sum_{i=0}^{k-1} \mathbb{E}_{\Phi^1, \cdots, \Phi^t \sim \mathcal{P}} \left[ \mathbb{E}_{s \sim \mathcal{P}_i^\pi} \left[ f^t \left( \mathrm{dom}(\psi_s) \cup \pi(\psi_s), \Phi^1, \cdots, \Phi^t \right) - f^t \left( \mathrm{dom}(\psi_s), \Phi^1, \cdots, \Phi^t \right) \right] \right]$$

$$= \sum_{i=0}^{k-1} \mathbb{E}_{\Phi^1, \cdots, \Phi^t \sim \mathcal{P}} \left[ \mathbb{E}_{\Phi \sim \mathcal{P}} \left[ f^t \left( V(\pi, \Phi)_{:i} \cup V(\pi, \Phi)_{i+1}, \Phi^1, \cdots, \Phi^t \right) - f^t \left( V(\pi, \Phi)_{:i}, \Phi^1, \cdots, \Phi^t \right) \right] \right]$$

$$= \mathbb{E}_{\Phi \sim \mathcal{P}} \left[ \mathbb{E}_{\Phi^1, \cdots, \Phi^t \sim \mathcal{P}} \left[ \sum_{i=0}^{k-1} \left( f^t \left( V(\pi, \Phi)_{:i} \cup V(\pi, \Phi)_{i+1}, \Phi^1, \cdots, \Phi^t \right) - f^t \left( V(\pi, \Phi)_{:i}, \Phi^1, \cdots, \Phi^t \right) \right) \right] \right]$$

$$= \mathbb{E}_{\Phi \sim \mathcal{P}} \left[ \mathbb{E}_{\Phi^1, \cdots, \Phi^t \sim \mathcal{P}} \left[ f^t(V(\pi, \Phi), \Phi^1, \cdots, \Phi^t) \right] \right]$$

$$= \mathop{\mathbb{E}}_{\Phi \sim \mathcal{P}} \left[ \sigma^t (V(\pi, \Phi)) \right]$$

$$= \sigma^t (\mathcal{W}(\pi)).$$

The third equality above is because the distribution of $\mathrm{dom}(\psi_s)$ and $\pi(\psi_s)$ with $s \sim \mathcal{P}_i^\pi$ is exactly the same as the distribution of $V(\pi, \Phi)_{:i}$ and $V(\pi, \Phi)_{i+1}$ with $\Phi \sim \mathcal{P}$. Note that this $\Phi$ is independent of $\Phi^1, \cdots, \Phi^t$. The last equality is because the distribution of $V(\pi, \Phi)$ with $\Phi \sim \mathcal{P}$ is exactly the distribution of the seed sets taken from the leaves of $\mathcal{T}(\pi)$, which exactly corresponds to the random-walk non-adaptive policy $\mathcal{W}(\pi)$. $\qquad\square$

**Lemma 2.** *Let $S = \mathrm{dom}(\psi^1)$ and $S^+ = S \cup \{u\}$ for any partial realization $\psi^1$ and any $u \notin \mathrm{dom}(\psi^1)$. Then we have*

$$\mathop{\mathbb{E}}_{\Phi^1, \Phi^2, \Phi^3 \sim \mathcal{P}} \left[ f\left( S^+, (\Phi_S^1 \cup \Phi_S^2 \cup \Phi_S^3, \Phi_u^1 \cup \Phi_u^2, \Phi_{V \setminus S^+}^1) \right) \right.$$
$$\left. - f\left( S, (\Phi_S^1 \cup \Phi_S^2 \cup \Phi_S^3, \Phi_{V \setminus S}^1) \right) \mid \Phi^1 \sim \psi^1 \right] \le \Delta_{f^2}(u \mid S). \tag{6}$$

$$\mathop{\mathbb{E}}_{\Phi^1, \Phi^2, \Phi^3 \sim \mathcal{P}} \left[ f\left( S^+, (\Phi_S^1 \cup \Phi_S^2 \cup \Phi_S^3, \Phi_u^1 \cup \Phi_u^2 \cup \Phi_u^3, \Phi_{V \setminus S^+}^1) \right) \right.$$
$$\left. - f\left( S^+, (\Phi_S^1 \cup \Phi_S^2 \cup \Phi_S^3, \Phi_u^1 \cup \Phi_u^2, \Phi_{V \setminus S^+}^1) \right) \mid \Phi^1 \sim \psi^1 \right] \le \Delta_{f^2}(u \mid S). \tag{7}$$

*Proof.* We first prove Inequality (6). To do so, we first expand the RHS of Eq. (6),

$$\Delta_{f^2}(u \mid S) = \mathop{\mathbb{E}}_{\Phi^2, \Phi^3 \sim \mathcal{P}} \left[ f^2\left( S^+, \Phi^2, \Phi^3 \right) - f^2\left( S, \Phi^2, \Phi^3 \right) \right]$$

$$= \mathop{\mathbb{E}}_{\Phi^2, \Phi^3 \sim \mathcal{P}} \left[ f\left( S^+, (\Phi_S^2 \cup \Phi_S^3, \Phi_u^2 \cup \Phi_u^3, \Phi_{V \setminus S^+}^2) \right) - f\left( S, (\Phi_S^2 \cup \Phi_S^3, \Phi_u^2, \Phi_{V \setminus S^+}^2) \right) \right]$$

$$= \mathop{\mathbb{E}}_{\Phi_S^2, \Phi_S^3 \sim \mathcal{P}_S} \left[ \mathop{\mathbb{E}}_{\Phi_u^2, \Phi_u^3 \in \mathcal{P}_u} \left[ \mathop{\mathbb{E}}_{\Phi_{V \setminus S^+}^2 \sim \mathcal{P}_{V \setminus S^+}} \left[ f\left( S^+, (\Phi_S^2 \cup \Phi_S^3, \Phi_u^2 \cup \Phi_u^3, \Phi_{V \setminus S^+}^2) \right) - \right. \right. \right.$$
$$\left. \left. \left. f\left( S, (\Phi_S^2 \cup \Phi_S^3, \Phi_u^2, \Phi_{V \setminus S^+}^2) \right) \right] \right] \right]. \tag{9}$$

The third equality above holds because $\Phi_S^2, \Phi_S^3, \Phi_u^2, \Phi_u^3, \Phi_{V \setminus S^+}^2, \Phi_{V \setminus S^+}^3$ are mutually independent, and $\Phi_{V \setminus S^+}^3$ does not appear inside the expectation term. Next, we expand the LHS of Eq. (6),

LHS of Eq. (6)

$$= \mathop{\mathbb{E}}_{\Phi_S^1, \Phi_S^2, \Phi_S^3 \sim \mathcal{P}_S} \left[ \mathop{\mathbb{E}}_{\Phi_u^1, \Phi_u^2 \in \mathcal{P}_u} \left[ \mathop{\mathbb{E}}_{\Phi_{V \setminus S^+}^1 \sim \mathcal{P}_{V \setminus S^+}} \left[ f\left( S^+, (\Phi_S^1 \cup \Phi_S^2 \cup \Phi_S^3, \Phi_u^1 \cup \Phi_u^2, \Phi_{V \setminus S^+}^1) \right) \right. \right. \right.$$
$$\left. \left. \left. - f\left( S, (\Phi_S^1 \cup \Phi_S^2 \cup \Phi_S^3, \Phi_u^1, \Phi_{V \setminus S^+}^1) \right) \mid \Phi^1 \sim \psi^1 \right] \right] \right]$$

$$= \mathop{\mathbb{E}}_{\Phi_S^2, \Phi_S^3 \sim \mathcal{P}_S} \left[ \mathop{\mathbb{E}}_{\Phi_u^1, \Phi_u^2 \in \mathcal{P}_u} \left[ \mathop{\mathbb{E}}_{\Phi_{V \setminus S^+}^1 \sim \mathcal{P}_{V \setminus S^+}} \left[ f\left( S^+, (\psi^1 \cup \Phi_S^2 \cup \Phi_S^3, \Phi_u^1 \cup \Phi_u^2, \Phi_{V \setminus S^+}^1) \right) \right. \right. \right.$$
$$\left. \left. \left. - f\left( S, (\psi^1 \cup \Phi_S^2 \cup \Phi_S^3, \Phi_u^1, \Phi_{V \setminus S^+}^1) \right) \right] \right] \right].$$

$$= \mathop{\mathbb{E}}_{\Phi_S^2, \Phi_S^3 \sim \mathcal{P}_S} \left[ \mathop{\mathbb{E}}_{\Phi_u^1, \Phi_u^3 \in \mathcal{P}_u} \left[ \mathop{\mathbb{E}}_{\Phi_{V \setminus S^+}^2 \sim \mathcal{P}_{V \setminus S^+}} \left[ f\left( S^+, (\psi^1 \cup \Phi_S^2 \cup \Phi_S^3, \Phi_u^1 \cup \Phi_u^3, \Phi_{V \setminus S^+}^2) \right) \right. \right. \right.$$
$$\left. \left. \left. - f\left( S, (\psi^1 \cup \Phi_S^2 \cup \Phi_S^3, \Phi_u^1, \Phi_{V \setminus S^+}^2) \right) \right] \right] \right].$$

$$= \mathop{\mathbb{E}}_{\Phi_S^2, \Phi_S^3 \sim \mathcal{P}_S} \left[ \mathop{\mathbb{E}}_{\Phi_u^2, \Phi_u^3 \in \mathcal{P}_u} \left[ \mathop{\mathbb{E}}_{\Phi_{V \setminus S^+}^2 \sim \mathcal{P}_{V \setminus S^+}} \left[ f\left( S^+, (\psi^1 \cup \Phi_S^2 \cup \Phi_S^3, \Phi_u^2 \cup \Phi_u^3, \Phi_{V \setminus S^+}^2) \right) \right. \right. \right.$$

$$- f\left(S,(\psi^1 \cup \Phi_S^2 \cup \Phi_S^3, \Phi_u^2, \Phi_{V\setminus S^+}^2)\right)\bigg]\bigg]\bigg].\tag{10}$$

The first equality above holds because all these random variables are independent. The second equality above holds because $\Phi_S^1 = \psi^1$ implied by $\Phi^1 \sim \psi^1$. In the third equality, we replace $\Phi_{V\setminus S^+}^1$ with $\Phi_{V\setminus S^+}^2$ and replace $\Phi_u^2$ with $\Phi_u^3$, because they follow the same probability distributions and are independent to the other distributions. In the last equality, we replace $\Phi_u^1$ with $\Phi_u^2$.

Comparing Eq. (9) and Eq. (10), we know that it suffices to prove that for any fixed partial realizations $\phi_S^2, \phi_S^3, \phi_u^2, \phi_u^3, \phi_{V\setminus S}^2$,

$$f\left(S^+, (\psi^1 \cup \phi_S^2 \cup \phi_S^3, \phi_u^2 \cup \phi_u^3, \phi_{V\setminus S^+}^2)\right) - f\left(S,(\psi^1 \cup \phi_S^2 \cup \phi_S^3, \phi_u^2, \phi_{V\setminus S^+}^2)\right)$$

$$\leq f\left(S^+, (\phi_S^2 \cup \phi_S^3, \phi_u^2 \cup \phi_u^3, \phi_{V\setminus S^+}^2)\right) - f\left(S,(\phi_S^2 \cup \phi_S^3, \phi_u^2, \phi_{V\setminus S^+}^2)\right).\tag{11}$$

Consider any node $v \in \Gamma(S^+, (\psi^1 \cup \phi_S^2 \cup \phi_S^3, \phi_u^2 \cup \phi_u^3, \phi_{V\setminus S^+}^2))\setminus\Gamma(S, (\psi^1 \cup \phi_S^2 \cup \phi_S^3, \phi_u^2, \phi_{V\setminus S^+}^2))$, we have the following observations: (1) under the realization $(\psi^1 \cup \phi_S^2 \cup \phi_S^3, \phi_u^2, \phi_{V\setminus S^+}^2)$ (or equivalently its live-edge graph), node $v$ cannot be reached from nodes in $S$; and (2) under the realization $(\psi^1 \cup \phi_S^2 \cup \phi_S^3, \phi_u^2 \cup \phi_u^3, \phi_{V\setminus S^+}^2)$ (or equivalently its live-edge graph), node $v$ can be reached via a path $P$ originated from node $u$, and $P$ does not contain any node in $S$.

Now, we are going to prove that $v \in \Gamma(S^+, (\phi_S^2 \cup \phi_S^3, \phi_u^2 \cup \phi_u^3, \phi_{V\setminus S^+}^2))\setminus\Gamma(S, (\phi_S^2 \cup \phi_S^3, \phi_u^2, \phi_{V\setminus S^+}^2))$. Since the path $P$ does not contain any node in $S$, we know that path $P$ also exists under the realization $(\phi_S^2 \cup \phi_S^3, \phi_u^2 \cup \phi_u^3, \phi_{V\setminus S^+}^2)$, i.e., node $v$ can be reached from node $u$ under realization $(\phi_S^2 \cup \phi_S^3, \phi_u^2 \cup \phi_u^3, \phi_{V\setminus S^+}^2)$. Moreover, we know that the realization $((\phi_S^2 \cup \phi_S^3, \phi_u^2, \phi_{V\setminus S^+}^2)$ has less live edges than the realization $(\psi^1 \cup \phi_S^2 \cup \phi_S^3, \phi_u^2, \phi_{V\setminus S^+}^2)$, so node $v$ can not be reached from set $S$ under the realization $(\phi_S^2 \cup \phi_S^3, \phi_u^2, \phi_{V\setminus S^+}^2)$. As a result, we have proved

$$\Gamma\left(S^+, (\psi^1 \cup \phi_S^2 \cup \phi_S^3, \phi_u^2 \cup \phi_u^3, \phi_{V\setminus S^+}^2)\right) \setminus \Gamma\left(S, (\psi^1 \cup \phi_S^2 \cup \phi_S^3, \phi_u^2, \phi_{V\setminus S^+}^2)\right)$$

$$\subseteq \Gamma\left(S^+, (\phi_S^2 \cup \phi_S^3, \phi_u^2 \cup \phi_u^3, \phi_{V\setminus S^+}^2)\right) \setminus \Gamma\left(S, (\phi_S^2 \cup \phi_S^3, \phi_u^2, \phi_{V\setminus S^+}^2)\right).\tag{12}$$

This proves Eq. (11) and thus concludes the proof of Inequality (6). Note that the above proof on Eq. (11) resembles the proof of submodularity of influence utility function $f$ on a live-edge graph, but Eq. (11) is a bit more complicated because it is on different live-edge graphs.

Next we prove the Inequality (7). Again, we first expand the RHS of Eq. (7).

$$\Delta_{f^2}(u \mid S) = \mathbb{E}_{\Phi_S^2,\Phi_S^3 \sim \mathcal{P}_S}\left[\mathbb{E}_{\Phi_u^2,\Phi_u^3 \in \mathcal{P}_u}\left[\mathbb{E}_{\Phi_{V\setminus S^+}^2 \sim \mathcal{P}_{V\setminus S^+}}\left[f\left(S^+,(\Phi_S^2 \cup \Phi_S^3, \Phi_u^2 \cup \Phi_u^3, \Phi_{V\setminus S^+}^2)\right)\right.\right.\right.$$

$$\left.-f\left(S,(\Phi_S^2 \cup \Phi_S^3, \Phi_u^2, \Phi_{V\setminus S^+}^2)\right)\bigg]\bigg]\bigg]$$

$$\geq \mathbb{E}_{\Phi_S^2,\Phi_S^3 \sim \mathcal{P}_S}\left[\mathbb{E}_{\Phi_u^2,\Phi_u^3 \in \mathcal{P}_u}\left[\mathbb{E}_{\Phi_{V\setminus S^+}^2 \sim \mathcal{P}_{V\setminus S^+}}\left[f\left(S^+,(\Phi_S^2 \cup \Phi_S^3, \Phi_u^2 \cup \Phi_u^3, \Phi_{V\setminus S^+}^2)\right)\right.\right.\right.$$

$$\left.-f\left(S^+,(\Phi_S^2 \cup \Phi_S^3, \Phi_u^2, \Phi_{V\setminus S^+}^2)\right)\bigg]\bigg]\bigg].\tag{13}$$

The inequality above is by the monotonicity of $f(S,\phi)$ on $S$. Next, we expand the LHS of Eq. (7). LHS of Eq. (7)

$$= \mathbb{E}_{\Phi_S^1,\Phi_S^2,\Phi_S^3 \sim \mathcal{P}_S}\left[\mathbb{E}_{\Phi_u^1,\Phi_u^2,\Phi_u^3 \in \mathcal{P}_u}\left[\mathbb{E}_{\Phi_{V\setminus S^+}^1 \sim \mathcal{P}_{V\setminus S^+}}\left[f\left(S^+,(\Phi_S^1 \cup \Phi_S^2 \cup \Phi_S^3, \Phi_u^1 \cup \Phi_u^2 \cup \Phi_u^3, \Phi_{V\setminus S^+}^1)\right)\right.\right.\right.$$

$$\left. - f\left(S^+,(\Phi_S^1 \cup \Phi_S^2 \cup \Phi_S^3, \Phi_u^1 \cup \Phi_u^2, \Phi_{V\setminus S^+}^1)\right) \mid \Phi^1 \sim \psi^1\bigg]\bigg]\bigg]$$

$$= \mathbb{E}_{\Phi_S^2,\Phi_S^3 \sim \mathcal{P}_S}\left[\mathbb{E}_{\Phi_u^1,\Phi_u^2,\Phi_u^3 \in \mathcal{P}_u}\left[\mathbb{E}_{\Phi_{V\setminus S^+}^1 \sim \mathcal{P}_{V\setminus S^+}}\left[f\left(S^+,(\psi^1 \cup \Phi_S^2 \cup \Phi_S^3, \Phi_u^1 \cup \Phi_u^2 \cup \Phi_u^3, \Phi_{V\setminus S^+}^1)\right)\right.\right.\right.$$

$$- f\left(S^+, (\psi^1 \cup \Phi_S^2 \cup \Phi_S^3, \Phi_u^1 \cup \Phi_u^2, \Phi_{V\setminus S^+}^1)\right)\Big]\Big]\Big].$$

$$= \mathop{\mathbb{E}}_{\Phi_S^2, \Phi_S^3 \sim \mathcal{P}_S}\left[\mathop{\mathbb{E}}_{\Phi_u^1, \Phi_u^2, \Phi_u^3 \in \mathcal{P}_u}\left[\mathop{\mathbb{E}}_{\Phi_{V\setminus S^+}^2 \sim \mathcal{P}_{V\setminus S^+}}\left[f\left(S^+, (\psi^1 \cup \Phi_S^2 \cup \Phi_S^3, \Phi_u^1 \cup \Phi_u^2 \cup \Phi_u^3, \Phi_{V\setminus S^+}^2)\right)\right.\right.\right.$$

$$\left.\left.\left. - f\left(S^+, (\psi^1 \cup \Phi_S^2 \cup \Phi_S^3, \Phi_u^1 \cup \Phi_u^2, \Phi_{V\setminus S^+}^2)\right)\right]\Big]\right]. \tag{14}$$

The last equality holds by replacing $\Phi_{V\setminus S^+}^1$ with $\Phi_{V\setminus S^+}^2$, because both have the same distributions and are independent from the other distributions. Similar to the previous lemma, comparing Eq. (13) and Eq. (14), it suffices to prove that for fixed partial realizations $\phi_S^2, \phi_S^3, \phi_u^1, \phi_u^2, \phi_u^3$ and $\phi_{V\setminus S^+}^2$,

$$f\left(S^+, (\psi^1 \cup \phi_S^2 \cup \phi_S^3, \phi_u^1 \cup \phi_u^2 \cup \phi_u^3, \phi_{V\setminus S^+}^2)\right) - f\left(S^+, (\psi^1 \cup \phi_S^2 \cup \phi_S^3, \phi_u^1 \cup \phi_u^2, \phi_{V\setminus S^+}^2)\right)$$

$$\leq f\left(S^+, (\phi_S^2 \cup \phi_S^3, \phi_u^2 \cup \phi_u^3, \phi_{V\setminus S^+}^2)\right) - f\left(S^+, (\phi_S^2 \cup \phi_S^3, \phi_u^2, \phi_{V\setminus S^+}^2)\right). \tag{15}$$

Consider any node $v \in \Gamma(S^+, (\psi^1 \cup \phi_S^2 \cup \phi_S^3, \phi_u^1 \cup \phi_u^2 \cup \phi_u^3, \phi_{V\setminus S^+}^2))\setminus\Gamma(S^+, (\psi^1 \cup \phi_S^2 \cup \phi_S^3, \phi_u^1 \cup \phi_u^2, \phi_{V\setminus S^+}^2))$, we have the following observations: (1) Node $v$ cannot be reached from any node in set $S^+$ under the realization $(\psi^1 \cup \phi_S^2 \cup \phi_S^3, \phi_u^1 \cup \phi_u^2, \phi_{V\setminus S^+}^2)$; and (2) node $v$ can be reached via a simple path $P$ originated from node $u$ under the realization $(\psi^1 \cup \phi_S^2 \cup \phi_S^3, \phi_u^1 \cup \phi_u^2 \cup \phi_u^3, \phi_{V\setminus S^+}^2)$, and $P$ does not contain any node in $S$ and any edge in $\phi_u^1 \cup \phi_u^2$.

Now, we prove that $v \in \Gamma(S^+, (\phi_S^2 \cup \phi_S^3, \phi_u^2 \cup \phi_u^3, \phi_{V\setminus S^+}^2))\setminus\Gamma(S^+, (\phi_S^2 \cup \phi_S^3, \phi_u^2, \phi_{V\setminus S^+}^2))$. Since path $P$ does not contain any node in $S$ and any edge in $\phi_u^1$, we know that path $P$ also exists under realization $(\phi_S^2 \cup \phi_S^3, \phi_u^2 \cup \phi_u^3, \phi_{V\setminus S^+}^2)$, i.e., node $v$ can be reached from node $u$ under realization $(\phi_S^2 \cup \phi_S^3, \phi_u^2 \cup \phi_u^3, \phi_{V\setminus S^+}^2)$. Moreover, we know that the realization $(\phi_S^2 \cup \phi_S^3, \phi_u^2, \phi_{V\setminus S^+}^2)$ has less live edges than the realization $(\psi^1 \cup \phi_S^2 \cup \phi_S^3, \phi_u^1 \cup \phi_u^2, \phi_{V\setminus S^+}^2)$, thus node $v$ cannot be reached from the set $S^+$ under realization $(\phi_S^2 \cup \phi_S^3, \phi_u^2, \phi_{V\setminus S^+}^2)$. Thus we can conclude that $v \in \Gamma(S^+, (\phi_S^2 \cup \phi_S^3, \phi_u^2 \cup \phi_u^3, \phi_{V\setminus S^+}^2))\setminus\Gamma(S^+, (\phi_S^2 \cup \phi_S^3, \phi_u^2, \phi_{V\setminus S^+}^2))$, this leads to Eq. (15) and concludes the proof of Inequality (7). $\qquad\square$

**Lemma 3.** *For any partial realization $\psi^1$ and node $u \notin \mathrm{dom}(\psi^1)$, we have*

$$\Delta_{f^3}(u \mid \psi^1) \leq 2\Delta_{f^2}(u \mid \mathrm{dom}(\psi^1)). \tag{8}$$

*Proof.* Again, for ease of notation, we set $S = \mathrm{dom}(\psi^1)$ and $S^+ = \mathrm{dom}(\psi^1) \cup \{u\}$, then we have

$$\Delta_{f^3}(u \mid \psi^1) = \mathop{\mathbb{E}}_{\Phi^1, \Phi^2, \Phi^3 \sim \mathcal{P}}\left[f\left(S^+, (\Phi_S^1 \cup \Phi_S^2 \cup \Phi_S^3, \Phi_u^1 \cup \Phi_u^2 \cup \Phi_u^3, \Phi_{V\setminus S^+}^1)\right)\right.$$

$$\left. - f\left(S, (\Phi_S^1 \cup \Phi_S^2 \cup \Phi_S^3, \Phi_{V\setminus S}^1)\right) \mid \Phi^1 \sim \psi^1\right]$$

$$= \mathop{\mathbb{E}}_{\Phi^1, \Phi^2, \Phi^3 \sim \mathcal{P}}\left[f\left(S^+, (\Phi_S^1 \cup \Phi_S^2 \cup \Phi_S^3, \Phi_u^1 \cup \Phi_u^2 \cup \Phi_u^3, \Phi_{V\setminus S^+}^1)\right)\right.$$

$$\left. - f\left(S^+, (\Phi_S^1 \cup \Phi_S^2 \cup \Phi_S^3, \Phi_u^1 \cup \Phi_u^2, \Phi_{V\setminus S^+}^1)\right) \mid \Phi^1 \sim \psi^1\right]$$

$$+ \mathop{\mathbb{E}}_{\Phi^1, \Phi^2, \Phi^3 \sim \mathcal{P}}\left[f\left(S^+, (\Phi_S^1 \cup \Phi_S^2 \cup \Phi_S^3, \Phi_u^1 \cup \Phi_u^2, \Phi_{V\setminus S^+}^1)\right) - f\left(S, (\Phi_S^1 \cup \Phi_S^2 \cup \Phi_S^3, \Phi_{V\setminus S}^1)\right) \mid \Phi^1 \sim \psi^1\right]$$

$$\leq \Delta_{f^2}(u|S) + \Delta_{f^2}(u|S) = 2\Delta_{f^2}(u|\mathrm{dom}(\psi^1)). \tag{16}$$

The inequality above is a direct consequence of Lemmas 2. $\qquad\square$

**Lemma 4.** *For any $t \geq 1$ and any subset $S \subseteq V$, $\sigma^t(S) \leq t \cdot \sigma(S)$.*

*Proof.* We have

$$\sigma^t(S) = \mathop{\mathbb{E}}_{\Phi^1, \cdots, \Phi^t \sim \mathcal{P}}\left[f^t(S, \Phi^1, \cdots, \Phi^t)\right] = \mathop{\mathbb{E}}_{\Phi^1, \cdots, \Phi^t \sim \mathcal{P}}\left[f\left(S, (\cup_{i \in [t]}\Phi_S^i, \Phi_{V\setminus S}^1)\right)\right]$$

$$= \underset{\Phi^1_{V\setminus S}\sim\mathcal{P}_{V\setminus S}}{\mathbb{E}}\left[\underset{\Phi^1_S,\cdots,\Phi^t_S\sim\mathcal{P}_S}{\mathbb{E}}\left[f\left(S,(\cup_{i\in[t]}\Phi^i_S,\Phi^1_{V\setminus S}))\right)\right]\right]. \tag{17}$$

We want to show that for any fixed $\phi^1_{V\setminus S}$,

$$\underset{\Phi^1_S,\cdots,\Phi^t_S\sim\mathcal{P}_S}{\mathbb{E}}\left[f\left(S,(\cup_{i\in[t]}\Phi^i_S,\phi^1_{V\setminus S}))\right)\right] \le \sum_{i\in[t]}\underset{\Phi^i_S}{\mathbb{E}}\left[f\left(S,(\Phi^i_S,\phi^1_{V\setminus S}))\right)\right]. \tag{18}$$

Once Eq.(18) is shown, we can combine with Eq.(17) to obtain

$$\sigma^t(S) \le \underset{\Phi^1_{V\setminus S}\sim\mathcal{P}}{\mathbb{E}}\left[\sum_{i\in[t]}\underset{\Phi^i_S}{\mathbb{E}}\left[f\left(S,(\Phi^i_S,\Phi^1_{V\setminus S}))\right)\right]\right]$$

$$= \sum_{i\in[t]}\underset{\Phi^1_{V\setminus S}\sim\mathcal{P}}{\mathbb{E}}\left[\underset{\Phi^i_S}{\mathbb{E}}\left[f\left(S,(\Phi^i_S,\Phi^1_{V\setminus S}))\right)\right]\right]$$

$$= \sum_{i\in[t]}\underset{\Phi^1\sim\mathcal{P}}{\mathbb{E}}\left[f(S,\Phi^1)\right] = t\cdot\sigma(S).$$

Thus the lemma holds. Now we prove Inequality (18). To do so, we fix partial realizations $\phi^1_S,\cdots,\phi^t_S$. If node $v\in\Gamma(S,\cup_{i\in[t]}\phi^i_S,\phi^1_{V\setminus S}))$, then we conclude that under the realization $(\cup_{i\in[t]}\phi^i_S,\phi^1_{V\setminus S})$, node $v$ can be reached via a path $P$ originated from some node $u\in S$, and only the starting node of $P$ is in $S$ and all remaining nodes in $P$ are not from $S$. Suppose in path $P$, the edge leaving node $u$ is contained in edge set $\phi^i_u$ for some $i\in[t]$. Then we conclude that node $v\in\Gamma(S,(\phi^i_S,\phi^1_{V\setminus S}))$, since the path $P$ exists under the realization $(\phi^i_S,\phi^1_{V\setminus S})$. This shows that $\Gamma(S,(\cup_{i\in[t]}\phi^i_S,\phi^1_{V\setminus S}))\subseteq \cup_{i\in[t]}\Gamma(S,(\phi^i_S,\phi^1_{V\setminus S}))$, which is sufficient to prove Inequality (18). $\qquad\square$

# B   Missing Proof of Section 3.2, Adaptivity Lower Bound

**Theorem 2.** *Under the IC model with myopic feedback, the adaptivity gap for the influence maximization problem is at least $e/(e-1)$.*

*Proof.* Consider the following construction for the influence graph: the influence graph $G = (L,R,E,p)$ is a bipartite graph with $|L| = \binom{m^3}{m^2}$ and $|R| = m^3$. All edges $(u,v)\in E$ are directed from the left part $L$ to the right part $R$, associated with probability $1/m$. More specifically, for any subset $X\subseteq R$ with size $m^2$, there is a node $u_X\in L$ such that the outgoing edges of $u_X$ are exactly $(u_X,v)$ for every $v\in X$. Thus the out-degree of every vertex in $L$ is $m^2$.

We first describe the main idea of the proof. The budget for the IM problem is $m^2$, i.e., we are allowed to select no more than $m^2$ seeds, and we would consider $m$ to be a very large number here. Intuitively, the expected number of nodes in $R$ that is reachable for a single node $u\in L$ is $m^2\cdot(1/m) = m$, and the influence spread is concentrated on its expected value for large $m$. In an adaptive solution, we could always make the expected marginal gain for the node we select equals the expected influence spread of a single node in $L$, by selecting nodes in $L$ such that none of its out-neighbors has been reached so far, unless there are too few nodes in $R$ that are not reachable. Since $m^2\cdot m = m^3$, the seeds we select would reach almost all but except $o(m^3)$ nodes in $R$, thus the influence spread of the adaptive policy is roughly $m^3$. While for a non-adaptive policy, it can select at most $m^2$ nodes from $L$ and for each node in $R$, on average, it is connected with at most $m^2\cdot m^2/m^3 = m$ seeds in $L$, we can easily prove that it is indeed the best allocation of seeds in $L$, and the expected probability for nodes in $R$ to be reached is $1-(1-1/m)^m\approx 1-1/e$. Moreover, since we are allowed to select no more than $m^2$ seeds in $R$ and they would not reach any other node, the contribution of this part is negligible. Thus the expected influence spread for the optimal non-adaptive solution would not exceed $(1-1/e)m^3$ and the adaptivity gap is $e/(e-1)$ on this graph.

The following two claims would make the above intuition formal.

**Claim 2.** *For any $\epsilon > 0$, when $m$ is large enough, we have $\mathrm{OPT}_A(G,m^2)\ge (1-\epsilon)m^3$.*

*Proof.* For any fixed $\epsilon > 0$, we would take $m$ such that $m \geq 48/\epsilon^2 \log m$. Consider the following adaptive policy $\pi$, which only selects nodes from the left part $L$. Moreover, for every node $u \in L$ selected by $\pi$, at the time of selection, none of $u$'s out-neighbors in $R$ has been reached yet from nodes selected by $\pi$ so far (this condition can be verified by an adaptive policy with myopic feedback). When there does not exist such node or the size of the seed set already equals to the budget, $\pi$ would stop. For $i \in \{1, \cdots, (1 - \epsilon/2)m^2\}$, let $\mathcal{E}_i$ denote the event that after selecting the $i$-th seed in $L$, the marginal gain of the influence spread is between $[(1 - \epsilon/2)m + 1, (1 + \epsilon/2)m + 1]$. We would give a lower bound on the conditional probability $\Pr[\mathcal{E}_i \mid \mathcal{E}_1, \cdots, \mathcal{E}_{i-1}]$. Under the condition $\cup_{j=1}^{i-1} \mathcal{E}_j$, the current influence spread on the right part $R$ is less than $(1 + \epsilon/2)m \cdot (1 - \epsilon/2)m^2 = (1 - \epsilon^2/4)m^3 < m^3 - m^2$, thus policy $\pi$ would not stop by now. Thus the marginal gain is the summation of $m^2$ independent binomial variables with mean $m$. By the Chernoff bound we have

$$\Pr[\mathcal{E}_i \mid \mathcal{E}_1, \cdots, \mathcal{E}_{i-1}] \geq 1 - exp(-\epsilon^2 m/12) \geq 1 - \frac{1}{m^3}. \tag{19}$$

Consequently,

$$\Pr[\cup_{i=1}^t \mathcal{E}_i] = \Pi_{i=1}^t \Pr[\mathcal{E}_i \mid \mathcal{E}_1, \cdots, \mathcal{E}_{i-1}] \geq (1 - \frac{1}{m^3})^{m^2} \geq 1 - \frac{1}{m^3} \cdot m^2 = 1 - \frac{1}{m}. \tag{20}$$

Thus the expected influence is greater than $(1 - \frac{1}{m}) \cdot (1 - \epsilon/2)m \cdot (1 - \epsilon/2)m^2 \geq (1 - \epsilon)m^3$. $\quad\square$

**Claim 3.** $\mathrm{OPT}_N(G, m^2) \leq (1 - (1 - 1/m)^m)m^3 + 2m^2$.

*Proof.* Let $S_L$ (resp. $S_R$) denote the seed set selected by the optimal non-adaptive policy from the left part $L$ (resp. right part $R$). For any node $u_i \in R$ where $i \in [m^3]$, let $x_i$ denote the number of $u_i$'s in-neighbors in the seed set $S_L$. Since the out-degree for each node in $S_L$ is $m^2$, we have $\sum_{i \in [m^3]} x_i \leq |S_L| \cdot m^2$ and the average number of in-neighbors is at most $|S_L| \cdot m^2/m^3 = |S_L|/m$. Furthermore, we can calculate the influence spread of $S_L$,

$$\sigma(S_L) = |S_L| + \sum_{i \in [m^3]} \Pr[u_i \text{ is reachable}]$$

$$= |S_L| + \sum_{i \in [m^3]} \left(1 - \left(1 - \frac{1}{m}\right)^{x_i}\right)$$

$$\leq |S_L| + m^3 \cdot \left(1 - \left(1 - \frac{1}{m}\right)^{|S_L|/m}\right)$$

$$\leq m^2 + m^3 \cdot \left(1 - \left(1 - \frac{1}{m}\right)^m\right). \tag{21}$$

The first inequality holds because function $g(x) = (1 - (1 - \frac{1}{m})^x)$ is concave. The last inequality holds because $|S_L| \leq m^2$. Now we have

$$\mathrm{OPT}_N(G, m^2) = \max_{\substack{S_L \subseteq L, S_R \subseteq R, \\ |S_L| + |S_R| \leq m^2}} \sigma(S_L \cup S_R) \leq \max_{\substack{S_L \subseteq L, \\ |S_L| \leq m^2}} \sigma(S_L) + \max_{\substack{S_R \subseteq L, \\ |S_R| \leq m^2}} \sigma(S_R)$$

$$\leq m^2 + m^3 \cdot \left(1 - \left(1 - \frac{1}{m}\right)^m\right) + m^2 = \left(1 - \left(1 - \frac{1}{m}\right)^m\right) \cdot m^3 + 2m^2. \tag{22}$$

This concludes the proof. $\quad\square$

Combining Claims 2 and 3, we can conclude that for any $\epsilon > 0$, there exists large enough $m$ such that $\mathrm{OPT}_A(G, m^2)/\mathrm{OPT}_N(G, m^2) \geq e/(e - 1) - \epsilon$. Letting $\epsilon \to 0$, we obtain the theorem. $\quad\square$

# C  Missing Proofs in Section 4

For the proofs in this section, let $\mathrm{Greedy}_N(G, k)$ (resp. $\mathrm{Greedy}_A(G, k)$) denote the influence spread for the non-adaptive greedy algorithm (resp. adaptive influence spread for the adaptive greedy algorithm), on the influence graph $G$ with a budget $k$.

The proof of Theorem 3 is complete once we prove the following lemma.

**Lemma 5.** *Adaptive greedy is $(1 - 1/e)$ approximate to the optimal non-adaptive policy.*

*Proof.* For a fixed influence graph $G$, let $S$ ($|S| = k$) denote the seed set selected by the optimal non-adaptive algorithm, where $s_i$ denotes the $i^{th}$ element in set $S$. We use $\mathcal{A}$ to denote adaptive greedy and for any $t \in \{0, 1, \cdots, k\}$, we use $U(t)$ to denote the expected adaptive influence spread of nodes selected by $\mathcal{A}$ in the first $i$ rounds, i.e.,

$$U(t) := \mathop{\mathbb{E}}_{\Phi \sim \mathcal{P}} \left[ f\left( V(\mathcal{A}, \Phi)_{:t}, \Phi \right) \right], \tag{23}$$

From the above definition, we can see that $U(0) = 0$ and $U(k) = \sigma(\mathcal{A})$. By Lemma 1, we have

$$U(t) = \sum_{i=0}^{t-1} \mathop{\mathbb{E}}_{s \sim \mathcal{P}_i^{\mathcal{A}}} \left[ \Delta_f \left( \mathcal{A}(\psi_s) \mid \psi_s \right) \right]. \tag{24}$$

Now, for any $t \in \{0, 1, \cdots, k-1\}$

$$
\begin{aligned}
U(t+1) - U(t) &= \mathop{\mathbb{E}}_{s \sim \mathcal{P}_t^{\mathcal{A}}} \left[ \Delta_f \left( \mathcal{A}(\psi_s) \mid \psi_s \right) \right] \\
&\geq \frac{1}{k} \sum_{i=1}^{k} \mathop{\mathbb{E}}_{s \sim \mathcal{P}_t^{\mathcal{A}}} \left[ \Delta_f \left( s_i \mid \psi_s \right) \right] \\
&= \frac{1}{k} \sum_{i=1}^{k} \mathop{\mathbb{E}}_{s \sim \mathcal{P}_t^{\mathcal{A}}} \left[ \mathop{\mathbb{E}}_{\Phi \sim \mathcal{P}} \left[ f\left( \mathrm{dom}(\psi_s) \cup \{s_i\}, \Phi \right) - f\left( \mathrm{dom}(\psi_s), \Phi \right) | \Phi \sim \psi_s \right] \right] \\
&= \frac{1}{k} \mathop{\mathbb{E}}_{s \sim \mathcal{P}_t^{\mathcal{A}}} \left[ \mathop{\mathbb{E}}_{\Phi \sim \mathcal{P}} \left[ \sum_{i=1}^{k} \left( f\left( \mathrm{dom}(\psi_s) \cup \{s_i\}, \Phi \right) - f\left( \mathrm{dom}(\psi_s), \Phi \right) \right) | \Phi \sim \psi_s \right] \right] \\
&\geq \frac{1}{k} \mathop{\mathbb{E}}_{s \sim \mathcal{P}_t^{\mathcal{A}}} \left[ \mathop{\mathbb{E}}_{\Phi \sim \mathcal{P}} \left[ f(\mathrm{dom}(\psi_s) \cup S, \Phi) - f(\mathrm{dom}(\psi_s), \Phi) | \Phi \sim \psi_s \right] \right] \\
&\geq \frac{1}{k} \mathop{\mathbb{E}}_{s \sim \mathcal{P}_t^{\mathcal{A}}} \left[ \mathop{\mathbb{E}}_{\Phi \sim \mathcal{P}} \left[ f(S, \Phi) - f(\mathrm{dom}(\psi_s), \Phi) | \Phi \sim \psi_s \right] \right] \\
&= \frac{1}{k} \left( \sigma(S) - U(t) \right). \tag{25}
\end{aligned}
$$

The first inequality holds since adaptive greedy $\mathcal{A}$ chooses the node that maximizes the expected marginal gain, i.e., for any partial realization $\psi$, $\Delta_f(\mathcal{A}(\psi) \mid \psi) \geq \Delta_f(s_i \mid \psi)$ for any $i \in [k]$. The second inequality is because the influence utility function $f(\cdot, \Phi)$ is submodular under a fixed realization $\Phi$. The third inequality holds because the influence utility function $f(\cdot, \Phi)$ is monotone under a fixed realization $\Phi$. The last equality utilizes the law of total expectation.

Now via standard argument, Eq. (25) implies that

$$
\begin{aligned}
\mathrm{Greedy}_A(G, k) = U(k) &\geq \left( 1 - \left( 1 - \frac{1}{k} \right)^k \right) \sigma(S) = \left( 1 - \left( 1 - \frac{1}{k} \right)^k \right) \mathrm{OPT}_N(G, k) \\
&\geq \left( 1 - \frac{1}{e} \right) \cdot \mathrm{OPT}_N(G, k). \tag{26}
\end{aligned}
$$

This concludes the proof. $\square$

We now prove Theorem 4. We first present a example showing that the non-adaptive greedy achieves at most $\frac{e^2+1}{(e+1)^2}$ approximation ratio.

**Lemma 6.** *Non-adaptive greedy algorithm has ratio at most $\frac{e^2+1}{(e+1)^2}$ with respect to the optimal adaptive solution, in the IC model with myopic feedback.*

*Proof.* Consider the following influence graph $G(V, E, p)$, where $V = V_1 \bigcup V_2 \bigcup V_3$, $|V_1| = d - 1$, $|V_2| = d$ and $|V_3| = 2d$. We would use $v_j^i$ to denote the $j^{th}$ node in $V_i$. Nodes in $V_1$ and $V_2$ have

unit weight while nodes in $V_3$ have weight $w$. Note that we could achieve the weight of $w$ by simply replacing each node with a chain of $w$ nodes with edge probability 1, so that as long as the head of the chain is activated, the whole chain is activated. There are directed edges from $V_1$ to $V_2$ and from $V_2$ to $V_3$. More specifically, for any $j \in [d], l \in [d-1]$, there is a direct edge from the node $v_l^1$ to the node $v_j^2$, associated with probability $1/d$. The node $v_j^2$ is connected to node $v_{2j-1}^3$ and $v_{2j}^3$, with probability $e/(e+1)$. The budget $k = \frac{e+3}{e+1}d$. We first consider the optimal adaptive solution and we observe that the optimal adaptive strategy can reach almost all nodes in $V_3$.

**Claim 4.** *For any $\epsilon > 0$, if we set $d \geq 2\log(2/\epsilon)/\epsilon^2$, then we have* $\mathrm{OPT}_A(G, k) \geq (1-\epsilon) \cdot 2dw$

*Proof.* Consider the following adaptive strategy: we first select all $d$ nodes in $V_2$ and observe which nodes in $V_3$ have not yet been reached, this can be done with myopic feedback. We would then use the left budget to select nodes in $V_3$ that have not been reached. Let $X_j = \mathbb{I}\{v_j^3 \text{ not activated by seed nodes in } V_2\}$ for $j \in [2d]$, where $\mathbb{I}\{\}$ is the indicator function. $X_j$'s are independent Bernoulli random variables with $\mathbb{E}[X_j] = \frac{1}{e+1}$. Then by the Chernoff bound,

$$\Pr[X_1 + \cdots + X_{2d} > \frac{2}{e+1}d + \epsilon d] \leq e^{-\frac{\epsilon d \cdot \epsilon(e+1)/2}{3}} \leq e^{-d\epsilon^2/2} \leq \frac{\epsilon}{2}. \qquad (27)$$

Consequently, the expected number of nodes in $V_3$ that have not been activated by seeds in $V_2$ is at most $\frac{\epsilon}{2} \cdot 2d + (1 - \frac{\epsilon}{2}) \cdot (\frac{2}{e+1}d + \epsilon d) \leq \frac{2}{e+1}d + 2\epsilon d$. But the adaptive greedy algorithm still has a budget of $\frac{2}{e+1}d$ to directly activate nodes in $V_3$, and thus the expected final number of non-activated nodes in $V_3$ is at most $2\epsilon d$. Thus we conclude the proof. $\qquad\square$

Next, we consider the greedy algorithm and have the following conclusion.

**Claim 5.** *The non-adaptive greedy algorithm would first select all $d-1$ nodes in $V_1$, and then select $\frac{2}{e+1}d + 1$ nodes in $V_2$. Consequently, we have that*

$$\mathrm{Greedy}_N(G, k) = (d-1) + \left[\left(\frac{2}{e+1}d + 1\right) + \left(1 - \left(1 - \frac{1}{d}\right)^{d-1}\right) \cdot \left(\frac{e-1}{e+1}d - 1\right)\right] \cdot (1 + \frac{2e}{e+1}w),$$
$$(28)$$

*when $d, w \to \infty$, we know that $\frac{\mathrm{Greedy}_N(G,k)}{dw} \to \frac{2e^2+2}{(e+1)^2}$.*

*Proof.* We first prove that greedy would first select all $d-1$ nodes in $V_1$. Consider that the greedy algorithm has already selected $j$ nodes in $V_1$ as seeds, with $j = 0, 1, \ldots, d-1$. Let $p_j$ denote the probability that a node in $V_2$ is activated in this case. We know that $p_j = 1 - (1 - \frac{1}{d})^j$. At this point, we know that the marginal gain for selecting the $(j+1)$-th node in $V_1$ is

$$M_1 = 1 + d \cdot \frac{1}{d}(1 - p_j) \cdot (1 + \frac{2e}{e+1}w) = 1 + (1 - p_j)(1 + \frac{2e}{e+1}w). \qquad (29)$$

In contrast, the marginal gain for selecting the first node in $V_2$ as a seed is

$$M_2 = (1 - p_j)(1 + \frac{2e}{e+1}w), \qquad (30)$$

and the marginal gain for selecting the first node in $V_3$ as a seed is

$$M_3 = p_j(1 - \frac{e}{e+1})w + (1 - p_j)w = \left(p_j \cdot \frac{1}{e+1} + (1 - p_j)\right)w. \qquad (31)$$

Therefore $M_1 > M_2$. Comparing $M_1$ with $M_3$, we use the fact that for all $j < d, p_j \leq 1 - 1/e$, and thus

$$M_1 - M_3 = 1 + (1 - p_j)(1 + \frac{2e}{e+1}w) - \left(p_j \cdot \frac{1}{e+1} + (1 - p_j)\right)w$$
$$> (1 - p_j)\frac{2e}{e+1}w - \left(p_j \cdot \frac{1}{e+1} + (1 - p_j)\right)w$$

$$= \left( \frac{e-1}{e+1} - \frac{e}{e+1} p_j \right) w$$

$$\geq 0. \tag{32}$$

Thus we conclude that greedy would select all $(d-1)$ nodes in $V_1$ first. Afterwards, we compare the marginal gain of selecting a node in $V_2$ versus selecting a node in $V_3$. Notice that if we select a node in $V_3$, we would definitely not select a node whose in-neighbor in $V_2$ is already selected as a seed, because it only decreases the marginal. Therefore, the marginal gains of selecting a node in $V_2$ or a node in $V_3$ are still given us $M_2$ and $M_3$. Thus, the difference of marginal gain is

$$\begin{aligned} M_2 - M_3 &= (1 - p_{d-1})(1 + \frac{2e}{e+1} w) - \left( p_{d-1} \cdot \frac{1}{e+1} + (1 - p_{d-1}) \right) w \\ &> (1 - p_{d-1}) \frac{2e}{e+1} w - \left( p_{d-1} \cdot \frac{1}{e+1} + (1 - p_{d-1}) \right) w \\ &= \left( \frac{e-1}{e+1} - \frac{e}{e+1} p_{d-1} \right) w \\ &= \left( \frac{e-1}{e+1} - \frac{e}{e+1} \left( 1 - \left( 1 - \frac{1}{d} \right)^{d-1} \right) \right) w \\ &\geq 0. \tag{33} \end{aligned}$$

Thus the marginal gain for selecting nodes in $V_2$ is greater than nodes in $V_3$ and greedy would select $\frac{2}{e+1} d + 1$ nodes in $V_2$. All in all, the expected utility for greedy is

$$\mathrm{Greedy}_N(G, k) = (d-1) + \left[ \left( \frac{2}{e+1} d + 1 \right) + \left( 1 - \left( 1 - \frac{1}{d} \right)^{d-1} \right) \cdot \left( \frac{e-1}{e+1} d - 1 \right) \right] \cdot (1 + \frac{2e}{e+1} w). \tag{34}$$

and when $d, w \to \infty$, we know that $\frac{\mathrm{Greedy}_N(G,k)}{dw} \to \frac{2e^2+2}{(e+1)^2}$. $\qquad \square$

Combining Claim 5 and Claim 4, we conclude that when $d, w \to \infty$,

$$\frac{\mathrm{Greedy}_N(G, k)}{\mathrm{OPT}_A(G)} \to \frac{e^2 + 1}{(e+1)^2} \approx 0.606. \tag{35}$$

$\qquad \square$

We then assert that the approximation ratio of adaptive greedy is no better than greedy.

**Lemma 7.** *The approximation ratio for the non-adaptive greedy algorithm is no worse than the adaptive greedy algorithm, over all graphs.*

*Proof.* Fix an influence graph $G(V, E, p)$, and any $k \in [n]$. We use $c$ to denote the approximation ratio of greedy, i.e.,

$$c = \frac{\mathrm{Greedy}_N(G, k)}{\mathrm{OPT}_A(G, k)}.$$

We construct a family of graph $G(w)$ such that the approximation ratio for adaptive greedy is approaching to $c$ when $w \to \infty$. The influence graph $G(w)$ consists of two parts, $G_1$ and $G_2$. The graph $G_1$ has same nodes as $G$, but it does not contain any edges, while the graph $G_2$ is exactly the same as $G$, except that the weight for each node is multiplied by a factor of $w$. Notice that we can always assign integral weights $w$ to a node by connecting it to a directed chain of length $w - 1$. For any node $v \in G_1$, $v$ has exactly one outgoing edge, connecting to the corresponding node in $G_2$, the edge will be live with probability 1.

Now, consider adaptive greedy on $G(w)$ with the same budget. Our first observation is that adaptive greedy will never choose nodes from $G_2$. This is because if the corresponding node in $G_1$ has not been chosen, the marginal gain of choosing the node in $G_1$ is always larger by 1, and if it has already

been chosen, the marginal gain to choose the node in $G_2$ is 0. Consequently, the adaptive greedy algorithm would always choose nodes in $G_1$. However, because myopic feedback only provides one step feedback after seed selection, selecting a node in $G_1$ would only provide the activation of its corresponding node in $G_2$ as the feedback, but this is already known for sure, and thus we do not get any useful feedback under myopic feedback model on this graph. Therefore, the adaptive greedy algorithm in this case behaves exactly the same as the non-adaptive greedy algorithm on the influence graph $G$, and the performance for adaptive greedy is

$$\text{Greedy}_A(G(w), k) = w \cdot \text{Greedy}_N(G, k) + k \leq (w + 1) \cdot \text{Greedy}_N(G, k). \tag{36}$$

Consider the optimal adaptive policy, a feasible adaptive policy is to ignore nodes in graph $G_1$ and perform the optimal adaptive policy on graph $G_2$, we have

$$\text{OPT}_A(G(w), k) \geq \text{OPT}_A(G_2(w), k) = w \cdot \text{OPT}_A(G, k). \tag{37}$$

By Eq. (36) and Eq. (37), the approximation ratio of adaptive greedy can be bounded as

$$\frac{\text{Greedy}_A(G(t), k)}{\text{OPT}_A(G(t), k)} \leq \frac{(w + 1) \cdot \text{Greedy}_N(G, k))}{w \cdot \text{OPT}_A(G, k)} = \frac{w + 1}{w} \cdot c \to c, \text{ when } w \to \infty. \tag{38}$$

This concludes the proof. □