[Reviews · NeurIPS 2019]

Reviewer 1



This paper studies an important problem, adaptive influence maximization, in social network analysis. This problem was first formally considered by Golovin and Krause but very few theoretical results have been found since then. The results provided in this paper are solid and may have a significant impact on this area. First, the upper bound of the adaptive gap is the first result of this kind concerning the adaptive influence maximization problem. As a corollary, the approximation ratio of the simple adaptive greedy algorithm under the myopic model can be confirmed as at least 0.25*(1-1/e), which, as mentioned in the paper, solves the conjecture of Golovin and Krause. The proof is obtained based on a novel construction of an intermediate seeding strategy. Second, the author also provides the lower bound based on sophisticated construction, which, more importantly, opens a question of how to fill the gap between the upper and lower bound. Finally, the authors provide the upper bound of the approximation ratio of the greedy strategy under the myopic feedback model. The proof indicates that myopic feedback does not help adaptive greedy, or from another perspective, adaptivity does not fundamentally useful under myopic feedback. The results here immediately lead future research to investigate better feedback models. I have thoroughly checked all the proofs and to my best believe they are sound. Some minors: line 214 page 5, may need rephrasing. line 242 page 5, one pair of brackets missed line 260 page 6, phi^1 might be phi Eq 6 page 7, the notation for a realization uses a generalized version of that defined in line 270 page 6. The authors may define this explicitly. line 469 page 13, "lemmas" might be "lemma" Scored Review: (Excellent Very Good Good Fair) Originality: Excellent Quality: Excellent Clarity: Very Good Significance: Very Good Update after Author response: The clarifications are clear.

Reviewer 2



The paper considers the adaptive influence maximization problem with myopic feedback under the independent cascade model. In the adaptive influence maximization problem, the seed nodes are sequentially selected one by one. After each seed selection, some information (feedback) about the spread of influence is returned to the algorithm which could be used for the selection of the next seeds. In the myopic feedback, the set of activated immediate neighbors of the selected seed is returned as such feedback. Finally, in the independent cascade model, every edge of the graph has an independent probability of passing the influence. This problem is shown to not be adaptive submodular. Therefore, it was an open problem if a constant approximation can be achieved. The paper shows that the answer is yes and prove that the greedy algorithm gives a (1-1/e)/4-approximation. The main idea of the proof is to show that for each adaptive policy, we could construct a non-adaptive randomized policy, such that the adaptive influence spread of the adaptive policy is at most 4 times the non-adaptive influence spread of the randomized policy. This shows that the adaptivity gap of the problem is at most 4 and proves the approximation factor of the greedy algorithm. The paper then proceeds to show that an adaptive version of the greedy algorithm also achieves the same approximation factor. They also show that there exist cases where the adaptive greedy and non-adaptive greedy (classic greedy) cannot find an approximation factor better than (e^2+1)/(e+1)^2. I personally am not a fan of this last result because it only focuses on the greedy algorithm. It would have been better to give an actual hardness of approximation result. I haven’t checked the proofs presented in the appendix but I believe the results are very interesting and the (1-1/e)/4-approximation result guarantees the acceptance. However as I mentioned before, I would really like to see a hardness of approximation result since the problem is not adaptive submodular and the hardness might be different than 1-1/e+epsilon.

Reviewer 3



The problem dealt with in this paper is the adaptive influence maximization with myopic feedback model. This problem was proposed by Golovin--Krause (2011), but they did not provide any bound on the approximation ratio of the adaptive greedy algorithm. They conjectured that this algorithm achieves constant-factor approximation. In this paper, the authors affirmatively solve this conjecture by analyzing the adaptivity gap. The adaptivity gap is the largest ratio of the objective values achieved by an optimal adaptive policy and an optimal non-adaptive policy among all graphs. The authors provide lower bound 1/4 and upper bound (1-1/e) on the adaptivity gap. By using this bound, the authors show both the adaptive and non-adaptive greedy algorithms are (1-1/e)/4-approximation to the optimal adaptive policy. Also, it is shown that neither the adaptive greedy nor the non-adaptive greedy is better than (e^2+1)/(e+1)^2-approximation. This paper is generally very well-written. In the proofs, this paper utilizes several novel ideas such as the multiplied version \sigma^t of the influence function. The open problem solved by this paper is significant not only in theory but also in practice. The results would give an impact on future research on influence maximization. I would be for the acceptance of this paper. Small comments: - In the first formula of (38), G(t) could be replaced with G(w). - In the proof of Claim 5, though it is not theoretically essential, 2d/(e+1) might not be an integer. It would be better to add a flooring function. - The statement of Lemma 7 would be a little bit misleading. In the current form, it sounds to me that "for any graph, the approximation ratio achieved by the non-adaptive greedy is no worse than the approximation ratio achieved by the adaptive greedy *in the same graph*." However, it is not what the authors mean. In my understanding, the authors mean that "for the approximation ratio of the non-adaptive greedy in any graph, there is some graph for which the approximation ratio of the adaptive greedy is equal to this one." Update after the author feedback: I have checked the author feedback. The authors explain why they did not conduct experiments, and I am satisfied with their comment. My opinion about the theoretical results remains the same, so I don't change my score.

[Author Response · NeurIPS 2019]

We would thank all reviewers for their insightful comments!

**Review #1:**

Thank you for the thoughtful comment.

We would consider adding a more detailed discussion in the camera-ready version if the paper is accepted. The main reason that our approach cannot be directly applied to other feedback models (like the full-adoption feedback model) or other information diffusion models (like linear threshold model) is that the feedback information in these models are no longer independent. To be more specific, in the full-adoption feedback model, the feedback information contains the full cascade of the selected node. The feedback for different nodes can share the status of same edges and thus the feedback are dependent. For the linear threshold model, we consider the case where the feedback information contains only whether nodes are activated or not. Even if we are restricted to myopic feedback, the feedback for different nodes are not independent since they can share common neighbors. To the best of our knowledge, the adaptivity gaps in these models are still open and all previous approaches failed to deal with dependent feedback information.

**Review #2:**

Thank you for the thoughtful comment.

We agree that showing a hardness results of $1 - 1/e - \varepsilon$ would be very interesting, if it is indeed the case. However, we believe that this task could be potentially very tough and thus is out of scope of the current paper. The toughness comes from both the formulation of the problem and the proof of the results. The original NP-hardness results of maximizing a submodular function is actually reduced via PCP theorem. We do not know whether there are any ''simple'' reductions can be used.

By the way, since most influence maximization algorithms are based on the greedy framework, our hardness result for greedy and adaptive greedy algorithms actually rules out a large number of existing algorithms.

**Review #3:**

Thank you for the thoughtful comment. For experiments, we will consider comparing adaptive greedy with non adaptive greedy, but this is not the main result of our paper. For the current study, our main purpose is to show the adaptivity gap, but due to the NP hardness we cannot get the optimal solution for any reasonably sized graph, so it is hard to validate our main theoretical results by experimental studies.

[Meta-Review · NeurIPS 2019]

This paper studies an adaptive version of the heavily studied influence maximization problem. The problem is the adaptive influence maximization problem with myopic feedback in the independent cascade model. In the adaptive influence maximization problem, the seed nodes are sequentially selected one by one. After each seed selection, some information (feedback) about the spread of influence is returned to the algorithm which could be used for the selection of the next seeds. In the myopic feedback, the set of activated immediate neighbors of the selected seed is returned as such feedback. Finally, in the independent cascade model, every edge of the graph has an independent probability of passing the influence. This problem was first formally considered by Golovin and Krause a decade ago. This problem was studied in the context of the adaptive submodularity framework. In this line of work one seeks objectives that are adaptive submodular which can then be optimized within a constant factor using a simple greedy algorithm. The main challenge with the model studied in this paper is that it is not adaptive submodular. It was an open problem if a constant approximation can be achieved. The paper gives a (1-1/e)/4-approximation algorithm for this problem as well as a lower bound.